



# Benchmarking GOCART-2G in the Goddard Earth Observing System (GEOS)

Allison B. Collow[1,2], Peter R. Colarco[3], Arlindo M. da Silva[2], Virginie Buchard[1,2], Huisheng Bian[1,3], Mian Chin[3], Sampa Das[3,4], Ravi Govindaraju[2,5], Dongchul Kim[1,3], and Valentina Aquila[6]

[1]University of Maryland Baltimore County, Baltimore, Maryland, US
[2]Global Modeling and Assimilation Office, NASA Goddard Space Flight Center, Greenbelt, Maryland, US
[3]Atmospheric Chemistry and Dynamics Lab, NASA Goddard Space Flight Center, Greenbelt, Maryland, US
[4] University of Maryland, College Park, MD
[5]Science Systems and Applications, Inc., Lanham, Maryland, US
[6]American University, Department of Environmental Science, Washington, District of Columbia, US

*Correspondence to*: Allison Collow (allison.collow@nasa.gov)

**Abstract.** The Goddard Chemistry Aerosol Radiation and Transport (GOCART) model, which controls the sources sinks and
chemistry within the Goddard Earth Observing System, recently underwent a major refactoring and update to the representation of physical processes. This paper serves to document code changes that were included in GOCART 2nd Generation (GOCART-2G) and establishes a benchmark simulation that is to be used for future development of the system. The code refactoring increases flexibility such multiple instances of an aerosol species can be run and interact with radiation and cloud microphysics, in addition to the output of multiple wavelength aerosol optical properties in support of data assimilation. From a science
perspective, a new radiatively active tracer, brown carbon, was added to distinguish smoke from other sources of organic aerosol thereby improving optical properties entering the radiative calculations. A four-year benchmark simulation was evaluated using in situ and space borne measurements to develop a baseline and prioritize future development. A comparison of simulated aerosol optical depth between GOCART-2G and MODIS retrievals indicates the model captures the overall spatial pattern and seasonal cycle of aerosol optical depth but overestimates aerosol extinction over dusty regions and underestimates
aerosol extinction over northern hemisphere boreal forests, requiring further tuning of emissions. This MODIS-based analysis is corroborated by comparisons to MISR and selected AERONET stations. Despite the underestimate of aerosol optical depth in biomass burning regions in GEOS, there is an overestimate in the surface mass of organic carbon in the United States, especially during the summer months.

## 1 Introduction

Aerosols are an important component of the atmosphere, with implications for air quality, cloud lifecycle, and the radiation budget. As general circulation models strive to take a comprehensive Earth-system approach, aerosol modules have become coupled to the atmosphere for use in numerical weather prediction (Colarco et al., 2010; Rémy et al., 2019), seasonal prediction (Molod et al., 2020), and reanalyses (Buchard et al., 2017; Randles et al., 2017), and have been shown to increase





forecast skill through changes in temperature (Bozzo et al., 2020). Aerosol modules handle the sources, sinks, and chemistry
within models; however, they vary in complexity and their diverse assumptions result in uncertainty and diversity in the
simulated aerosol life cycle and optical properties (Textor et al., 2006; Tsigaridis et al., 2014; Gliß et al, 2021). Modelled
aerosols are driven by emissions that are prescribed, as is the case for smoke and anthropogenic emissions, or calculated by
parameterizations coupled with meteorological fields, such as for sea salt and dust. Choices for which dataset or
parameterization to use can have a large impact on the modelled aerosol loading. For example, uncertainty arises for biomass
burning, where six commonly used emission datasets were found to differ by a factor of 3.8 (Pan et al., 2020), and dust optical
depth model diversity is dominated by the simulated dust source strength (Kim et al., 2019).

Aerosol modules also require assumptions regarding the number of aerosol species, size of aerosol particles, particle
size-dependent gravitational settling, and particle swelling in response to changes in relative humidity. Simpler aerosol bulk
aerosol models typically prescribe sizes and lack microphysical processes found in modal (eg. Liu et al., 2016) and sectional
schemes (eg. Yu et al., 2015). Among the global models included in the International Cooperative for Aerosol Prediction
(ICAP) multi-model ensemble (MME) study, there is variability in the species, particularly related to nitrate and biomass
burning smoke, as well as the number of size bins used to discretize the particle size distribution for a given species (Table 1
of Xian et al., 2019). The settling for each size bin can follow the same size-based parameterization for each bin or settle at
different rates depending on the size bin. While some aerosols are hydrophobic, others swell in the presence of water vapor,
which can affect particle settling speeds and optical properties. A large range in the scattering enhancement factor as a function
of relative humidity, f(RH), was found in a comparison of ten Earth system models (Burgos et al., 2020), which can result in
varying aerosol extinction even where aerosol loadings are similar.

A commonly used bulk aerosol module is the Goddard Chemistry Aerosol Radiation and Transport (GOCART)
module, which traces its origin to an offline aerosol transport model driven by the assimilated meteorological fields from the
Goddard Earth Observing System (GEOS; Chin et al. 2002, 2014). GOCART was later coupled to GEOS to enable short-term
aerosol forecasts and provide a platform for aerosol data assimilation (Colarco et al., 2010). It has also been implemented in
NOAA's Unified Forecast System (Lu et al., 2016, Zhang et al., 2022) and the Weather Research and Forecasting Model
(WRF). In its legacy form, GOCART has handled the sources, sinks, and chemistry of externally mixed aerosols within the
GEOS model and its individual systems. Near real-time aerosol forecasts began in GEOS Forward Processing in 2011 (Figure
S1), though GOCART had previously been used within GEOS for field campaign support. An aerosol analysis has been
produced retrospectively in reanalysis systems such as the Modern Era retrospective Analysis for Research and Applications,
Version 2 (MERRA-2, Randles et al. 2017, Buchard et al. 2017). Though GOCART has evolved over the past two decades, a
recent overhaul of the module has been completed to pave the way for future development.

This paper serves to document updates that have been implemented in GOCART since the production of MERRA-2,
including a suite of science changes and code improvements that encompass GOCART second generation (GOCART-2G). A
four-year simulation is then evaluated in Section 4 to benchmark the performance of GOGART-2G and provide a reference
for future development of aerosol modelling within GEOS. One major difference between the evaluation presented here



compared to prior evaluations of aerosols within GEOS from the MERRA-2 system (Randles et al. (2017); Buchard et al. 2017) is that here, aerosol optical depth (AOD) is not assimilated, like Colarco et al. (2010). While meteorology is constrained

in the benchmark simulation, no aerosol data assimilation is included, and aerosol distributions are governed solely by processes in the model. GOCART-2G is intended to be used in future versions of GEOS FP and reanalysis products with GEOS, hence the need for proper documentation and evaluation.

## 2 GOCART aerosol module in GEOS

### 2.1 Background

75       GOCART-2G includes seven radiatively active aerosol tracers that are considered externally mixed: sea salt, dust, organic carbon, brown carbon, black carbon, sulphate, and nitrate. Like in MERRA-2, sea salt (SS) and dust (DU) are comprised of five non-interacting size bins (Table A1). Sea salt emissions are based on Gong (2003), with some key modifications: 1) friction velocity is used instead of 10 m wind speed, which required tuning for the constants within the parameterization, and 2) addition of a correction term dependent on sea surface temperature, similar to the work of Jaegle et

al. (2011) but tuned to improve the agreement of simulated sea salt AOD with MODIS retrieved AOD. Default dust emissions follow Ginoux et al. (2001), see Table 1. The smallest size bin for dust is further divided into four sub-bins for optics calculations according to Tegen and Lacis (1996). Organic (OC), brown (BR), and black (BC) carbon have hydrophobic and hydrophilic components. Upon emission, 50% of organic carbon, 50% of brown carbon, and 80% of black carbon are considered hydrophobic (Chin et al., 2002) and transition to hydrophilic at a e-folding time scale of 2.5 days (Maria et al.,

2004). A factor of 1.8 is implemented upon emission to convert organic carbon, including the tracer for brown carbon, to particulate organic matter (POM), which has been increased from the factor of 1.4 used in MERRA-2 based on observations from recent airborne campaigns (Hodzic et al., 2020). Emission sources of carbonaceous aerosol include biomass burning, biogenic, and anthropogenic emissions. Biomass burning emissions are released uniformly throughout the planetary boundary layer (PBL) depth, while anthropogenic emissions enter only in the lowest model level. While the source data for biomass

burning emissions is consistent with MERRA-2, anthropogenic emissions now come from the Community Emissions Data System (CEDS) v_2021_04_21 (Table 1; Hoesly et al., 2018; doi: 10.5281/zenodo.4741285); currently CEDS emissions are available up to 2019. Additional information is provided in Section 2.2 pertaining to the implementation of brown carbon and secondary organic aerosol (SOA) as these tracers were added as part of GOCART-2G.

       A single tracer is used for the sulphate ion, $SO_4^{2-}$. Volcanic emissions of $SO_2$ are from Carn et al. (2017) with explosive

emissions updated through 2021 as of this writing, while biomass burning and anthropogenic emissions of $SO_2$ and $SO_4^{2-}$ are consistent with the carbon emissions (Table 1). Sulphate chemistry follows Chin et al. (2000) in which sulphate is formed from the oxidation of $SO_2$ and the precursor dimethyl sulfide (DMS) in the presence of hydroxide (OH) and $NO_3$ and aqueously via titration of hydrogen peroxide ($H_2O_2$). In a traditional GOCART-2G simulation, these oxidant fields are provided in archived



monthly data from previous full chemistry simulations and the diurnal cycle is imposed on the OH field. GOCART-2G can
also run interactively with a gas chemistry module in which these oxidant fields are updated at every time step.

Nitrate was incorporated into GEOS in 2017, after production began for MERRA-2, based on the approach used for
the Global Modeling Initiative (GMI) chemistry transport model (CTM) (Bian et al. 2017). Three particle size groups are
included for nitrate aerosol in GOCART-2G: a fine mode bin and two coarse mode bins (Table 1 in Bian et al., 2017). The
fine mode bin for nitrate is simulated using the thermodynamic equilibrium model Regional Particulate Model Aerosol
Reacting System (RPMARES) (Saxena et al., 1986) for the gas phase, aqueous chemical cycling of nitrate gas-aerosol
partitioning in a system of $SO_4^{2-} - NO_3^- - NH_4^+ - H_2O$ (Table 2 in Bian et al., 2017), and a first order heterogenous reaction
of $HNO_3$ on mineral dust and sea salt. The two coarse mode bins form from heterogenous production only. Additional tracers
are included for ammonia ($NH_3$) and the ammonium ion ($NH_4^+$) that are necessary for the $SO_4^{-2}$-$NO_3^-$-$NH_4^+$-$H_2O$ system.
Biomass burning, anthropogenic, and oceanic emissions of $NH_3$ prescribed from emission datasets (Table 2). Precursor gases
for sulphate and nitrate are prescribed based on a prior MERRA-2 replay coupled to the GMI stratosphere-troposphere
chemical mechanism (MERRA-2 GMI; Strode et al., 2019).

**Table 1. Summary of aerosol emissions in the GOCART-2G benchmark simulation. GOCART-2G can be run with differing emissions sources and dataset resolutions if desired.**

| Emission Type | Species | Source | Temporal Resolution | Spatial Resolution |
|---|---|---|---|---|
| **Anthropogenic (including ship and aircraft)** | OC, BC, SO$_2$, SO$_4$, NH$_3$ | CEDS (doi: 10.5281/zenodo.4741285) | Monthly | 0.5, downscaled to 0.15625 |
| **Biomass Burning** | BR, BC, SO$_2$, NH$_3$ | QFED v2.5r1 (Darmenov and da Silva, 2015) | Daily, with a fixed diurnal cycle based on latitude | 0.1 |
| **Volcanic** | SO$_2$ | Carn et al. 2017 | Daily Eruptive and Outgassing | Point-sources |
| **Dust** | DU | Wind driven (Ginoux et al., 2001) | Model | Model Resolution |
| **Sea Salt** | SS | Wind driven (Gong, 2003; Jaegle et al., 2011) | Model | Model Resolution |
| **Species prescribed for aerosol chemistry** | H$_2$O$_2$, OH, NO$_3$, HNO$_3$ | MERRA-2 GMI (Strode et al 2019) | Monthly | 0.5 x 0.625 |
| | DMS | Lana et al. (2011); Liss and Merlivat (1986) | Monthly | 0.5 x 0.625 |
| | Open Ocean NH$_3$ | Bouwman et al. (1997) | Monthly | 0.5 x 0.625 |


**Table 2. Summary of aerosol parameterizations in GOCART-2G**

| Function | Specie(s) | Parameterization |
|---|---|---|
| Boundary Layer Turbulent Mixing | All | Lock et al., 2000; Louis, 1979 |



| Moist Convection | All | Grell and Freitas, 2014 |
|---|---|---|
| Settling Velocity | All | Fuchs, 1964 |
| Dry Deposition | All | Wesely, 1989 |
| Wet Deposition | All | Giorgi and Chameides, 1986; Balkanski, et al. 1993; Liu et al., 2001 |
| Optical Properties | All | Hess et al., 1998; Colarco et al., 2014; Colarco et al., 2017 |
| Sulphate Chemistry | Sulphate | Chin et al. 2000 |
| Nitrate Chemistry | Nitrate | Bian et al. 2017; Saxena et al., 1986 |

Optics look up tables (LUTs) to convert from the simulated aerosol masses to optical quantities such as aerosol optical depth (AOD) are derived from Mie (spherical) calculations using parameters from the Optical Properties of Aerosols and Clouds
(OPAC; Hess et al., 1998) and as described in Chin et al. (2002) and Colarco et al. (2010), except for dust, which is based on Colarco et al. (2014), and for brown carbon (see below). Optical properties are a function of aerosol species, particle size, and relative humidity (except for dust which is assumed hydrophobic). From an optics perspective, hygroscopic growth occurs based on a specified growth factor as listed in the Appendices of Kemppinen et al. (2022). The resulting optics tables are available for download at https://portal.nccs.nasa.gov/datashare/iesa/aerosol/AerosolOptics/ (last access 5 December 2022)
and the versions used in the initial release of GOCART-2G are given in Table A2. These high spectral resolution tables are useful for computing diagnostic optical quantities like AOD and backscatter, as shown later. They are also available in an aggregated format to provide optical properties needed to compute aerosol forcing at the spectral bands used in the model's radiative transfer code, RRTMG (Clough et al., 2005; Iacono et al., 2008).

## 2.2 Updates to Aerosol Speciation

130       Three major changes with regards to aerosol speciation were implemented as part of GOCART-2G. Brown carbon was added as a new radiatively active sub-species of carbon. Secondary organic aerosol (SOA) is now used to form brown and organic carbon from volatile organic carbon (VOC). Finally, a mechanism to produce sulphate in the stratosphere has been added. Additional details pertaining to these updates are given in the remainder of this section.

### 2.2.1 Implementation of Primary Organic Aerosol

135       Beginning with GOCART-2G, a distinction is made between "non-absorbing" anthropogenic and "absorbing" (also referred to as "brown") biomass burning organic aerosol. Anthropogenic emissions of organic carbon, which are emitted based on CEDS, are solely considered to be "non-absorbing" organic carbon, with spectral optical properties that follow the OPAC database and are as in Chin et al. (2002) and Colarco et al. (2010). Biomass burning emissions of organic carbon from QFED are considered absorbing "brown" carbon and assigned optical properties that have spectrally varying absorption at





wavelengths shorter than 550 nm as described in Colarco et al. (2017). This distinction was found in Colarco et al. (2017) to improve the comparison of the absorbing aerosol-sensitive aerosol index between the model and data retrieved from the Ozone Monitoring Instrument (OMI) onboard the NASA Aura spacecraft. The optical properties between our absorbing and non-absorbing organic aerosol components are identical at wavelengths equal to and greater than 550 nm and treated by identical chemical and loss processes in the model.

**2.2.2 Implementation of Secondary Organic Aerosol**

A simplified SOA mechanism is employed that scales VOC emissions in terms of carbon monoxide (CO) emissions from anthropogenic and biomass sources. Following Kim et al. (2015) we assume production of anthropogenic VOC at a rate of 0.069 g (g CO)$^{-1}$ and biomass burning VOC at a rate of 0.013 g (g CO)$^{-1}$. The VOC tracers are advected and assumed to convert to SOA via reaction with the prescribed MERRA-2 GMI OH fields with a rate constant of $1.25\times10^{-11}$ cm$^3$ molecule$^{-1}$

sec$^{-1}$ (Hodzic and Jimenez, 2011). The SOA produced is apportioned to the hydrophilic modes of organic (anthropogenic) and brown (biomass burning) carbon. Biogenic aerosols, including isoprene, monoterpene, and other terpenes are provided from the Model of Emissions of Gases and Aerosols from Nature (MEGAN; Guenther et al., 2012) and enter the model through the Harvard–NASA Emission Component software (HEMCO, Keller et al., 2014) and are assigned to the hydrophilic mode of the organic carbon component. Unlike anthropogenic and biomass burning SOA that are produced in the air via the reaction of

VOCs and OH, biogenic SOA is produced by scaling MEGAN emitted VOCs at the point of emission.

**2.2.3 Implementation of Stratospheric Sulphate Aerosol**

An optional simplified stratospheric sulphate mechanism is implemented following the mechanism described in English et al. (2011). A tracer for carbonyl sulphide (OCS) is added to the model with a specified surface mixing ratio boundary condition of 490 ppt$_v$. OCS is largely inert in the troposphere and the model has been spun up so that a well-mixed distribution

is achieved. Photochemical destruction of OCS is managed by the stratospheric chemistry package StratChem described in Nielsen et al. (2017). The reactions considered include binary consumption of OCS by OH and atomic oxygen, O($^3$P), and photolytic destruction of OCS (the dominant process), using rate constants from Sander et al. (2011). The sulphur is assumed oxidized to SO$_2$ and then passed to GOCART, which computes the sulphate aerosol production using the same series of reactions as above for the tropospheric sulphate aerosol production. This mechanism provides us a simplified representation

of the naturally occurring background stratospheric sulphate. While this mechanism is included in the benchmark experiment analysed in Section 4, it is currently not intended for use in a typical model simulation such as what is used to produce GEOS FP.

**2.3 Code Refactoring**

A major refactoring of the GOCART source code was completed to improve performance, flexibility, and code quality

within GOCART-2G. GOCART has been split into its own repository (https://github.com/GEOS-ESM/GOCART.git) with specific low-level interfaces that do not depend on the Earth System Model Framework (ESMF, https://earthsystemmodeling.org) and are independent of the overall GEOS architecture. This allows for code to be effectively





shared with external organizations. Performance was enhanced through optimization of settling and nitrate chemistry parameterizations, eliminating extraneous calculations, and removing known bugs. The other code refactoring consisted of eliminating non-standard Fortran, eliminating redundant and legacy constructs, reducing duplicated logic within and across components, implementing cleaner component resource files, improving procedure and variable names in the source code to make the intent obvious to users and developers, and splitting large procedures with well-defined responsibilities.

A large component of the refactoring involved more widespread adoption of the ESMF within the parent GOCART-2G component. Improved flexibility within the code is essential for future development of GOCART-2G within GEOS. Carbon, sulphate, nitrate, sea salt, and dust, now have their own ESMF components and can instantiate multiple active and/or passive instances at one time; an active instance participates in the physical coupling with the host model. For example, carbonaceous aerosol is one of GOCART-2G's children and black, brown, and organic carbon are each run as an active instance of the carbonaceous aerosol component. Previously, GOCART provided aerosol optical properties at the specific bands required by the radiation package, but diagnostic file output was restricted to the 550 nm wavelength. To better support data assimilation of multi-wavelength aerosol data, the model is now able to directly output aerosol optical properties at multiple wavelengths without the need for an offline utility. The model is also able to output stratospheric AOD using the GEOS tropopause height.

## 3 Observational Datasets Used for Model Evaluation

### 3.1 Moderate Resolution Imaging Spectroradiometer (MODIS) Neural Net Retrieval (NNR)

Here we evaluate AOD at 550 nm in GOCART-2G using observations from Collection 6.1 of the Moderate Resolution Imaging Spectroradiometer (MODIS) aboard the Aqua satellite (Levy et al., 2015). The particular MODIS dataset used for this evaluation is the Neural Net Retrieval (NNR) described in Section 3.2.2 of Randles et al. (2016), which bias corrects and homogenizes MODIS observations to be consisted with AERONET. THE NNR algorithm relies on cloud-cleared, gas-corrected reflectances used by the Deep Blue (Sayer et al., 2019) and Dark Target (Remer et al., 2020) retrievals and uses a neural net trained on co-located AERONET direct sun AOD measurements. The monthly mean NNR AOD retrievals are obtained by a weighted average based on the number of pixels available for a given 0.25° latitude by 0.3125° longitude grid box. The same NNR-based analysis was also carried out for the Terra satellite, complemented by additional measurements from the Multi-angle Imaging SpectroRadiometer (MISR); these results are presented in the supplemental material. GEOS has been sampled such that model data is only included when and where MODIS observations are available at the three hourly timestep of the MODIS NNR product.

### 3.2 AERONET

The AErosol RObotic NETwork (AERONET) is a collection of ground-based stations equipped with Cimel sun photometers for measuring spectral sun irradiance and sky irradiances (Holben et al., 1998). Under cloud-free conditions, AOD





is computed as the total optical depth measured by the sun photometer minus the contribution from Rayleigh scattering and

trace gases. For comparison to GEOS, Version 3 of the Level 2 product, which includes cloud screening, is utilized (Giles et al., 2019). Although AERONET provides spectrally varying AOD, only AOD at 550 nm is examined in addition to the Angstrom exponent computed using 470 nm and 870 nm. For stations that do not report AOD at 550 nm, the Angstrom exponent for 440 nm and 675 nm is used to convert the AOD at 500 nm to 550 nm.

### 3.3 OMPS-LP

The Ozone Mapping and Profiler Suite (OMPS) aboard the Joint Polar Satellite System (JPSS) Suomi National Polar-orbiting Partnership (S-NPP) satellite contains a limb profiler (LP) that can observe aerosol with the stratosphere. Stratospheric AOD at 869 nm is evaluated using observations from OMPS-LP (Taha et al., 2021) to validate volcanic eruptions and pyrocumulonimbus (PyroCB) reaching the stratosphere. The data from OMPS LP are presented as the daily, zonal mean of the stratospheric AOD, evaluated by integrating the retrieved extinction from the GEOS-derived tropopause altitude to the 40

km top altitude of the OMPS LP retrievals. Data are not available during periods of instrument issues and under low/no-sun conditions (e.g., polar night). Although the algorithm includes cloud screening, some polar stratospheric clouds are evident in the dataset, as shown below.

### 3.4 CALIOP

Since 2006, the Cloud Aerosol Lidar with Orthogonal Polarization (CALIOP), aboard NASA's CALIPSO ATrain

satellite (Winker et al., 2007, 2009), has provided insights about aerosol vertical structure. For this study, the highest-quality lidar Level 1.5 standard data product version V1.00 was employed (NASA, 2019): a cloud-cleared dataset with a 20 km horizontal and 60 m vertical resolution for a height up to 20.2 km. The observations include contributions from both aerosols and gas molecules (Rayleigh scattering), so our analysis is limited to the total (aerosols + molecular) attenuated backscatter coefficient.

### 225  3.5 Surface Particulate Matter from IMPROVE and EMEP

Like in Buchard et al. (2016) and Provençal (2017), surface aerosol mass is evaluated over the United States using data         provided         by         the         Interagency         Monitoring         of         Protected         Visual         Environments         (IMPROVE, http://vista.cira.colostate.edu/Improve/) Program and over Europe using data from the European Monitoring and Evaluation Programme (EMEP, https://ebas.nilu.no/). IMPROVE and EMEP monitoring sites are typically located in rural areas

representative of the region and with minimal influence from localized urban pollution.  Following the module description for the IMPROVE network (Hand et al., 2011), $PM_{2.5}$ in GOCART-2G was computed using the equation below for aerosol with an aerodynamic diameter of 2.5 microns (Collow et al., 2023). Variable names in the equation are consistent with those given in the file specification document for GEOS Forward Processing (Lucchesi, 2018). The multiplication factors of 0.9614 for bin 1 of dust and 0.3871 for bin 3 of sea salt account for a conversion to aerodynamic diameter and the fact that only a portion





of the bin is smaller than 2.5 microns. Though not done here, other studies have used the entirety of bin 1 for dust in comparison to IMPROVE observations due to a wide range in the shape factor for dust (Kim et al., 2021).

Reconstructed PM$_{2.5}$ is given by

PM$_{2.5}$ = 0.9614*DU001 + f$_{ss,rh}$*(SS001+SS002+ 0.3871*SS003) +

      + OCPHOBIC + BCPHOBIC + BRPHOBIC + f$_{oc,rh}$ * OCPHILIC +

      + f$_{bc,rh}$ * BCPHILIC + f$_{br,rh}$ * BRPHILIC + f$_{su,rh}$ * SO4 + f$_{ni,rh}$ * NH4a + f$_{ni,rh}$ * NO3an1

where the growth factor with relative humidity, $f_{x,rh}$, for each species is calculated as

$$f_{x,rh} \; = \; 1 + \left( \left( \left( \frac{r_{rh}}{r_0} \right)^3 - 1 \right) \; x \; \frac{\rho_{Water}}{\rho_{Dry\;Species}} \right)$$

using the radius specified for a given relative humidity from the optics files listed in Table 4 as $r_{rh}$ and the radius at 0% relative humidity for $r_0$.

Following their respective documentations, PM$_{2.5}$ for the IMPROVE sites in the United States was computed using a
relative humidity of 35% (Hand et al., 2011) while PM$_{2.5}$ for the EMEP sites in Europe was computed using a relative humidity of 50%. GEOS was sampled according to when and where observations were available. Note that IMPROVE observations are collected every three days while data from EMEP ranges in temporal frequency from one hour to six days. EMEP observations are also not homogeneous with respect to the instruments and measurements of individual aerosol species at each site.

## 4 Evaluation of GOCART-2G

A benchmark simulation for GOCART-2G was carried out for the period of 2016 through 2019 using GEOS Release 10.23.0 (https://github.com/GEOS-ESM/GEOSgcm/releases/tag/v10.23.0) on a cubed-sphere c180 grid (~0.5° spatial resolution) with 72 vertical levels. Meteorology, particularly atmospheric temperature, specific humidity, and winds were replayed to the analysis from MERRA-2 (Gelaro et al., 2017), while boundary conditions for sea surface temperature and sea ice concentration were from the Reynolds analysis (Reynolds et al., 2002). Two pyroCb events were included in the simulation
for British Columbia in 2017 (Torres et al., 2020; Das et al., 2021) and Australia in late December 2019 (Schwartz et al., 2020). There is no assimilation of aerosol optical depth or observational constraint for aerosol extinction or mass. Therefore, all observations used for comparison are independent from the model simulation.





## 4.1 Aerosol Mass Budget

Emissions and production from each aerosol species are presented in Figures 1 and 2. Wind-driven dust is emitted
primarily over Saharan Africa, Saudi Arabia, the Asia deserts, the Simpson desert of Australia, and the southern tip of South
America (Figure 1a), in agreement with Colarco et al. (2010), Randles et al. (2017) and Rémy et al. (2019). The seasonal cycle
of dust emissions peaks in boreal spring and is minimized during the fall months (Figure 2a). Most dust is deposited near the
source regions, however there is notable transport of Saharan dust across the Atlantic Ocean (Figure 3a).

Sea salt emissions are enhanced along the northern and southern hemisphere storms tracks as well as the Intertropical
Convergence Zone (ITCZ), with little variability across the seasonal cycle. While the spatial pattern is similar, sea salt
emissions have decreased from MERRA-2 (Randles et al., 2017). In a correction since MERRA-2, sea salt is not emitted over
the Great Lakes or Caspian Sea. Most sea salt is emitted in the coarse mode, with the largest contribution from bin 3 (mode
radius of 2.4 µm). Relative to the largest three bins, emissions from bins 1 and 2 are negligible to the total mass (Figure 2b).
Nearly all sea salt is deposited over the ocean, in elevated quantities over the storm tracks and ITCZ (Figure 3b).

Carbonaceous aerosol is emitted over land (Figure 1b-d), with a seasonal cycle that peaks in the boreal summer due
to the temporal variability in biomass burning (Figure 2c-d). Anthropogenic emissions account for, on average, 62% of the
total black carbon emissions and 21% of the organic aerosol emissions. Brown carbon, emitted through biomass burning,
ranges from 37% to 65% of the monthly emissions of organic aerosol. There is also a contribution of brown carbon produced
from SOA.

Sulphate is directly emitted within GEOS from the anthropogenic emissions and has a contribution that is produced
from the oxidation of dimethyl sulphide (DMS), methane sulfonic acid (MSA), and sulphur dioxide ($SO_2$). Emission and
production of sulphate is maximized in densely populated areas including China, India, Europe, and the Eastern United States
(Figure 1e). Anthropogenic emission of $SO_2$ is the largest contribution to sulphate production and is responsible for the subtle
downward trend of sulphate production over the four-year timeseries. The summertime peaks in sulphate production during
2018 and 2019 are in response to the explosive volcanic eruptions of Kilauea in May 2018 and Raikoke in June 2019 (Figure
2e) while the broader summertime peaks in gaseous production of sulphate are associated with biomass burning emissions of
$SO_2$.

Nitrate aerosol is not directly emitted. Most nitrate forms in response to heterogenous production on dust and sea salt
aerosols (Figure 2f). A somewhat bimodal seasonal cycle in the production of nitrate occurs due to spring and fall peaks in the
emission of ammonia ($NH_3$). Due to the anthropogenic and agricultural nature of ammonia emissions, the spatial pattern of
nitrate deposition is very similar to that for organic and brown carbon. Most nitrate aerosol is deposited close to the source
while some is transported over the ocean by the atmospheric circulation (Figure 3f).





## 4.2 Comparison to Observational Datasets

### 4.2.1 Satellite Based Aerosol Optical Depth

A broad, global comparison of the AOD between MODIS and GEOS is shown in Figure 4. The model performs well over the ocean, however there are notable biases over land. AOD is too high in GEOS across northern Africa and Saudi Arabia, suggesting there could be too much dust in the model. Although smaller in magnitude, the overestimate in AOD extends into the central Atlantic due to transported dust. The positive bias in AOD is larger in magnitude when GEOS is compared to
MODIS aboard Terra relative to Aqua and suggests there could be a further issue with the diurnal cycle of dust emissions (Figure S2). However, this positive bias in AOD in dusty regions is not as large when GEOS is compared to MISR (Supplemental Figure S3) or AERONET (Figure 9, shown later). Conversely, a negative bias in AOD is present in the northern hemisphere boreal regions that is larger in magnitude when the comparison is made to Aqua.

Monthly mean timeseries of global mean AOD over ocean, in addition to the mean seasonal cycles, can be found in
Figure 5. In the top two panels, the solid black line represents the MODIS observations while the coloured shading accumulates the optical depth for each aerosol species in GEOS. Though difficult to see in the global spatial map, it is evident that AOD is underestimated in the model over the ocean. There is a seasonal cycle in the bias such that it is maximized during the months of March, September, and October and minimized during the boreal summer and winter (Figure 5d). MODIS indicates a bimodal seasonal cycle for total AOD, with one peak in the Northern Hemisphere late winter and early spring that is not present
in GEOS, and another during the summer that persists later into the season than in the model. The largest contribution to the total AOD comes from sea salt and in agreement with the fact that emissions cover a large fraction of the domain, there is little temporal variability in the optical depth for sea salt. Peaking in April, nitrate makes up the smallest contribution to the total AOD over the ocean. Peaks in sulphate are present in the boreal summers of 2018 and 2019, coincident with peaks in the gaseous production of sulphate due to large volcanic eruptions as shown in Figure 2e.

The analysis of AOD over land is broken down into eight continental scale regions. A spatial map demonstrating the geographic extent of each region is in the supplemental document (Figure S5). Beginning with North Africa in Figure 6a, the region is dominated by dust that typically peaks in the spring and summer months. GEOS can produce the observed temporal variability in AOD; however, the magnitude of AOD is higher than MODIS throughout the entire timeseries. This is likely due to an overestimate of dust emissions.

GEOS underestimates the AOD in South Asia, North America, South America, Siberia, and Europe (Figure 6d-h). The Americas, Siberia, and South Asia are influenced by biomass burning aerosol. Biomass burning aerosol is often underestimated in models, including GEOS, and in many cases have errors due to assumptions made for the particle properties (Zhong et al., 2022 and references within). Collow et al. (2022) demonstrated the GEOS struggles to match the observed mass extinction efficiency within a smoke plume. It is likely that the negative bias in these regions is in response to biomass burning
aerosol. Europe and South Asia are more complicated due to higher relative proportions of dust and sulphate. Dust emissions are tuned in GEOS using a global metric. It is therefore plausible that there are errors in the transport of dust to Europe and the





overall life cycle of dust from Asian deserts such as the Gobi and Taklamakan Deserts. A lack of the negative bias in AOD over South Asia in comparison to Terra indicates the underestimation of AOD in GEOS contains a diurnal cycle (Figure S6d).

GEOS completely misses the observed seasonal cycle in AOD over Europe. For this reason, Europe was further
divided into subregions as indicated by Figure 7 (See Figure S8 for the geographical depiction of the subregions). There is decent agreement in AOD between MODIS and GEOS over the Iberian Peninsula and Scandinavia. Conversely, GEOS does not capture the summertime maxima in AOD across central Europe or the United Kingdom. This will be further elaborated upon through a comparison with AERONET observations in Section 4.2.2 and an evaluation of surface mass in Section 4.2.4.

There is remarkable agreement in the AOD over South Africa and Australia with GEOS capturing the seasonal cycle
and magnitude from the observations (Figure 6b-c). The fact that South Africa is also dominated by biomass burning aerosol but does not have the negative bias seen in other biomass burning regions suggests there could be an overestimate of AOD due to another species or that the optical properties for brown and organic carbon in GEOS are better suited for the fuel types burned in Africa rather than the boreal forests of North America and Siberia and the rainforests of South America.

### 4.2.2 AERONET

Representative AERONET stations were selected for evaluation based on a comparison among dozens of stations in North America, Europe, and northern Africa. Due to the poor agreement in the seasonal cycle of AOD in Europe between GEOS and MODIS, Mainz, Germany was selected as the site demonstrates characteristics of others in the area. The AERONET site is adjacent to both rural and urban landscapes and is in a moderately to highly polluted region. In agreement with the
comparisons to MODIS, GEOS tends to underestimate the AOD with respect to the AERONET observations and has a mean negative bias of 0.28, in log space, that tends to be larger in magnitude during the summer months (Figure 8). In addition to smaller values of AOD occurring more frequently in GEOS compared to the observations, there is also less variability in the AOD. GEOS has a better agreement for the Angstrom Exponent, computed using 440 nm and 675 nm, accurately having the dominant aerosol in the fine mode. Potential reasons for the underestimate in AOD may be a lack of emissions from smaller
scale sources that are not represented in the CEDS dataset or insufficient biomass burning aerosol that is transported from North America.

Comparisons between GEOS and AERONET stations across northern Africa and Saudi Arabia are consistent with respect to the mean bias in the model relative to MODIS NNR. Tamanrasset was chosen for additional evaluation since it is in northern Africa where GEOS overestimates AOD compared to MODIS (Figure 9). The AERONET site is in the highlands of
the Algerian Sahara, away from industrial activity, making dust the primary aerosol species. Here, there is a positive mean bias in the modelled AOD of 0.18 and a reasonable correlation between GEOS and AERONET of R=0.84, computed using log(AOD+0.01) (Figure 9b). GEOS overestimates the AOD when the AERONET observations lie between 0.1 and 0.5 as demonstrated in Figure 9b. Agreement between the model and observations is not as good for the Angstrom exponent as the





correlation is only 0.48 and there is a mean bias of -0.15, indicating that aerosol in the model is often coarser than seen by
AERONET.

As shown in the comparison to MODIS, Southern Africa is dominated by biomass burning aerosol. Mongu, located in central south Africa within Zambia, was selected as a representative site for smoke. Despite good agreement between GEOS and MODIS on a continental scale for Southern Africa, there is considerable underestimation in AOD within the model when compared to AERONET at a local scale (Figure 10a and b). This is especially the case for the southern hemispheric winter
months when biomass burning is prevalent. The correlation of 0.85 at Mongu is on par with what was reported for the M2Replay, a MERRA-2 like simulation without the assimilation of AOD, in Randles et al. (2017). As shown by the kernel density estimate in Figure 10b, the correlation between the observations and GEOS is weaker for lower values of AOD. GEOS has a smaller amplitude in the Angstrom exponent such that there is an underestimate during the southern hemisphere summer months (Figure 10c). GEOS is likely correctly characterizing the July peaks in AOD as biomass burning aerosol but is missing
coarse mode aerosols, perhaps dust, during the warmer months.

The AERONET station in Langley, Virginia demonstrates behaviour typical of other stations and is close to the national average timeseries for AOD across the United States. Located on the southern tip of the Chesapeake Bay less that 40 km from Norfolk, Virginia, the Langley AERONET site often experiences urban and marine aerosol regimes, with occasional intrusions of smoke and dust. At this station, GEOS overestimates the lower values of AOD and underestimates the higher
values of AOD (Figure 11a), giving a poorer correlation than at the sites in Europe and the Sahel (Figure 11b). GEOS does not have as much variability in the Angstrom exponent as the observations but accurately represents that there is fine mode aerosol. A summary for 77 AERONET stations across the United States and Canada is given in the form of a kernel density estimate in Figure 12. Numerous stations underestimate AOD during the summer months, in agreement with the MODIS evaluation. Exceptions to this are stations in the desert southwest including Tucson, Flagstaff, Table Mountain CA, and USC,
where GEOS simulates higher AOD than AERONET. These stations are characterized by a measured AOD below national average (Figure S11).

### 4.2.3 OMPS-LP Stratospheric AOD

Newly added diagnostics in GOGART-2G include total aerosol scattering and extinction in the stratosphere, which allows for comparison to observations from OMPS-LP. Figure 13 shows the daily, zonal mean stratospheric AOD at 870 nm
from OMPS LP (panel a) and the GEOS simulation (panel b). GEOS modelled fields are masked where OMPS LP does not report retrievals either due to polar night conditions, scattering angle filtering, or missing data from spacecraft operations issues. Note some high AOD values along the northernmost points hugging the polar night line, particularly evident in January; these are unfiltered polar stratospheric cloud artifacts present in the OMPS LP data set (Ghassan Taha, personal communication) not included in the GEOS simulation. Generally, the model reproduces the observed seasonal variability and
magnitude of the stratospheric AOD and has markers for significant stratospheric perturbing events such as volcanic eruptions (Aoba in the tropics in 2018, Ulawun in the tropics in 2019, Raikoke at high northern latitudes in 2019) and pyrocumulonimbus



events (notably the British Columbia fires in high northern latitudes in late 2017). Even the seasonal variability evident exiting polar night is well captured in the model. Persistence of volcanic plumes following events however is not well captured in the model, suggesting difficulties with vertical placement and so long-range transport.

### 4.2.4 Vertical Profile of Attenuated Backscatter

To assess the vertical structure of aerosols in the GEOS-GOCART-2G model, we selected four regions of particular interest, as defined by Buchard et al. (2017). These included the dust transport region from northern Africa to the North Atlantic, the biomass burning regions of southern Africa and the Amazon, and an area over the continental United States. Figure 14 shows the June-July-August 2016 regional average of CALIOP 532 nm aerosol attenuated backscatter in black, and the corresponding attenuated backscatter sampled in space and time from GEOS-GOCART-2G in red (Supplemental figures S12-S15 show curtain plots of attenuated backscatter coefficients over the same regions). Generally, the GEOS-GOCART-2G attenuated backscatter profile tends to exhibit comparable vertical structure as CALIOP in all four regions of study. Notably, GEOS-GOCART-2G attenuated backscatter values agree well with CALIOP values within the CALIOP 25th-75th percentile range and their maximum values are located at around the same height.

As observed in our MERRA-2 study (Buchard et al., 2017), near-surface attenuated backscatter is underestimated relative to CALIOP in the Northern and Southern African regions, particularly for sea salt type aerosols near the ocean (Figures S14-S15). This could be due to either errors in the aerosol mass or in the hygroscopic growth assumption during the conversion from aerosol mass to optical properties. Nonetheless, calibration errors in CALIOP needs also to be considered as they tend to accumulate near the surface, making it difficult to place too much confidence in CALIOP values near the surface.

### 4.2.5 Surface Mass

Across the United States, surface particate matter is evaluated in GEOS relative to the IMPROVE network. GEOS overestimates $PM_{2.5}$ throughout the entire period of 2016 through 2019 however the model is well correlated to the observations (Figure 15). The 2017 and 2018 wildfire seasons were particularly bad in the United States as indicated by the summertime maxima in $PM_{2.5}$ in the IMPROVE observations and GEOS. The total fine surface matter is further divided into individual aerosol species in Figure 16. Like with the total $PM_{2.5}$, sulphate aerosol is consistently overestimated in GEOS. The IMPROVE observations indicate a seasonal cycle in sulphate that peaks in the summer, which is muted in GEOS. GEOS also struggles with the seasonal cycle for fine mode nitrate, overexaggerating the summertime minimum and wintertime maximum. The largest contributor to the overestimate of $PM_{2.5}$ in GEOS is organic carbon. During biomass burning events in the summers of 2017 and 2018, the mean surface concentration of organic matter in the model exceeds the mean plus one standard deviation in the observations. Although the sampling differs, AOD is underestimated with respect to satellite observations during the same events (Figure 6e). This indicates either too much aerosol is at the surface and not transported higher in the atmosphere and/or the mass extinction efficiency for smoke is too low in the model. Dust suffers from the opposite problem. Both the mean and the variability are underestimated by GEOS, with the largest bias during the summer months. Dust emissions were





tuned for more prominent regions such as the Sahara Desert. It is likely the emissions are not representative for the soil

conditions in the United States in addition to deficiencies in the long-range transport (Kim et al., 2019; Kim et al., 2021).

       The European Monitoring and Evaluation Programme (EMEP) had 67 stations across Europe with $PM_{2.5}$ data for the period of 2016 through 2019 however only a fraction of those also provided sulphate, nitrate, and carbon. There were no observations of dust available. Four representative stations within Germany and one in Poland have been selected due to their availability of data and consistency with instrumentation. GOCART-2G overestimates surface $PM_{2.5}$, especially during the

winter months (Figure 17). This is the opposite bias from Provençal et al. (2017) which evaluated the MERRAero reanalysis, and there are multiple reasons as to why there could be a larger aerosol concentration in the GOCART-2G simulations (which do not assimilated aerosol data). Aside from investigating a later time period for a subset of stations, nitrate and brown carbon were not included in MERRAero, although data assimilation may have apportioned the mass adjustments to the represented species. Additionally, we used an aerodynamic diameter for the particle size and accounted for hygroscopic growth since the

observations are acclimated to a relative humidity of 50% prior to being recorded, in contrast to the geometric diameter and assumption of dry aerosol used by Provençal et al. (2017). Relating the seasonal cycle of surface aerosol mass in Central Europe to the AOD in Figure 7d, there is an evident mismatch.

       To further diagnose potential contributions to positive bias in $PM_{2.5}$ over Europe, sulphate, nitrate, and carbon are evaluated. Like with $PM_{2.5}$, all species are overestimated by the model (Figure 18). Most easily seen by comparing the spread

between the $25^{th}$ and $75^{th}$ percentiles, GEOS captures the seasonal cycle of nitrate, organic carbon, and black carbon to some extent. The late winter peak in nitrate occurs a month two early in the model with a drastic decrease in the spring, perhaps indicating an issue with the emissions. While the seasonal cycle of carbonaceous aerosols is exaggerated in GOCART-2G, it correctly predicts a summertime minimum and a November maximum in black carbon. Given that PM2.5, sulphate, nitrate, and carbon are all overestimated in Europe, it is evident that there is a concern much larger than processes related to a single

species, as was the case for the United States. With only five stations analysed, representativeness becomes a concern when comparing a single point to a box with a resolution of roughly 50 km. However, the site description for Melpitz, one of the stations used, states that the site is representative of the Central European background troposphere following comparison with multiple other sites (https://gawsis.meteoswiss.ch/GAWSIS/#/search/station/stationReportDetails/0-20008-0-MEL, last accessed 24 February 2023). Other plausible explanations include biases in the modelled planetary boundary layer height and

aggressive hygroscopic growth to match a relative humidity of 50%.

## 5 Discussion

       GOCART, the underlying aerosol module within the Goddard Earth Observing System (GEOS) underwent an overhaul that coupled science changes with a code refactoring to enable future development of modelled aerosols within the system. Primary science changes focused on the introduction of the new radiatively active species, brown carbon, as well as



secondary organic aerosols. As part of the new species, MEGAN was coupled to HEMCO to provide biogenic emissions and the ratio of organic carbon to particulate organic matter was revised.

The modernization of GOCART-2G was necessary to enable future development. The use of multiple instances for a single species is employed for the three sub-species of carbon. This development could be expanded upon in future versions with, for example, ash as an additional instance of dust. The ability to have diagnostics provided in multiple user selected
wavelengths is particularly useful for aerosol assimilation and facilitates the comparison of the model with other sensors, such as OMPS LP. At the present time, GEOS assimilates AOD at 550 nm. It is anticipated that additional wavelengths will be added for aerosol assimilation after GEOS transitions to a Joint Effort for Data assimilation Integration (JEDI) based system. Assimilated information pertaining to the Angstrom exponent will be highly beneficial, giving the model a sense of the aerosol speciation from the observations.

In its current form, GOCART-2G can reproduce observed aerosol properties but has some notable potential for improvement. The spatial pattern of AOD across the globe is generally captured and the magnitude and seasonal cycle of AOD agrees well with MODIS satellite observations. Conversely, regions characterized by dust or biomass burning aerosols have overestimated and underestimated AOD, respectively. Further evaluation of surface aerosol mass in the United States suggests the mass extinction efficiency for biomass burning aerosol is too low in GOCART-2G. This is corroborated by evaluations of
GEOS with GOCART-2G using data collected from recent airborne field campaigns (Collow et al., 2022).

Though not discussed here, there are additional features of GOCART-2G that would benefit from future development. Bian et al. (2019) demonstrated a discrepancy in the particle size distribution for sea salt between GEOS and Particle Analysis by Laser Mass Spectrometry (PALMS) observations collected during the NASA ATom campaign. Although the largest two size bins for sea salt in GEOS were too coarse to be observed, it was evident that the model underestimated fine mode sea salt
in the first two size bins. There are multiple papers in the literature that evaluate dust. Yu et al. (2020) noted that GEOS underestimated emissions of dust from haboobs and did not loft dust high enough into the middle troposphere for sufficient transport, resulting in an underestimate of the dust AOD in the Caribbean during a substantial dust event in June 2020. It was also pointed out by Kramer et al. (2020) that transported dust is overabundant in the boundary layer and has a particle size that is too large. There is also room for improvement in aerosol transport. As an example, Das et al. (2017) showed the biomass
burning plume over the southeast Atlantic descends much too rapidly. With the flexibility and user-friendly refactored code implemented within GOCART-2G it is anticipated that aerosol model developers will be able to work together to progress GOCART such improvements can be seen within these features in future versions.

**Code Availability**

GEOS, including GOCART-2G, is a publicly available Earth System model with source code at https://github.com/GEOS-
ESM and https://doi.org/10.5281/zenodo.8059710. The archived code includes software to set up and run the model, compute AOD from MODIS Level 2 reflectances, and post process the model output.



**Data Availability**

All observational data used are from publicly available datasets. MODIS Level 2 reflectances are available from http://dx.doi.org/10.5067/MODIS/MOD04_L2.006 for Terra and http://dx.doi.org/10.5067/MODIS/MYD04_L2.006 for Aqua, CALIOP data can be downloaded at https://doi.org/10.5067/CALIOP/CALIPSO/LID_L15-STANDARD-V1-00, AERONET observations can be downloaded at https://aeronet.gsfc.nasa.gov/cgi-bin/webtool_aod_v3, IMPROVE data can be downloaded from the Federal Land Manager Environmental Database at

http://views.cira.colostate.edu/fed/DataWizard/Default.aspx, and EMEP data can be downloaded from EBAS at https://ebas-data.nilu.no/. Model data, in addition to the observational data used, is archived at http://dx.doi.org/10.5281/zenodo.8212822.

**Author Contribution**

AC, PC, and VB contributed to the visualization. PC, AdS, and RG contributed to the software and data curation. AC was responsible for original draft preparation and PC, AdS, VB, MC, HB, DK, SD, and VA contributed to review and editing.

**Competing Interests**

The authors declare that they have no conflict of interest.

**Acknowledgements**

This work was accomplished through computing resources from the NASA Center for Climate Simulation (NCCS). Elliot Sherman is acknowledged for his work in refactoring GOCART. The software infrastructure team at NASA's Global Modeling

Assimilation was invaluable throughout the process of developing GOCART-2G. We wish to thank Tom Clune, Ben Auer, Weiyuan Jiang, Matt Thompson, and Atanas Trayanov for their assistance, as well as Anton Darmenov for providing his expertise throughout the code refactoring.

We thank the AERONET PIs and Co-Is, and their staff, for establishing and maintaining the 80 sites used in this investigation.


IMPROVE is a collaborative association of state, tribal, and federal agencies, and international partners. US Environmental Protection Agency is the primary funding source, with contracting and research support from the National Park Service. The Air Quality Group at the University of California, Davis is the central analytical laboratory, with ion analysis provided by Research Triangle Institute, and carbon analysis provided by Desert Research Institute.



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

**Figures**









**Figure 1: Emissions of (a) dust, sea salt, (b) black carbon, (c) organic carbon, (d) brown carbon, and (e) sulphate as well as the production of (d) brown carbon from secondary organic aerosol, (e) sulphate, and (f) nitrate averaged for the period of January 2016 through December 2019 in the GEOS GOCART-2G benchmark simulation.**


**Figure 2: Timeseries of emissions and production of (a) dust, (b) sea salt, (c) black carbon, (d) organic carbon, brown carbon, (e) sulphate, and (f) nitrate for the period of January 2016 through December 2019 in the GEOS GOCART-2G benchmark simulation.**



**Figure 3: Deposition of (a) dust, sea salt, (b) black carbon, (c) organic carbon, (d) brown carbon, (e) sulphate, and (f) nitrate averaged**
**for the period of January 2016 through December 2019 in the GEOS GOCART-2G benchmark simulation.**

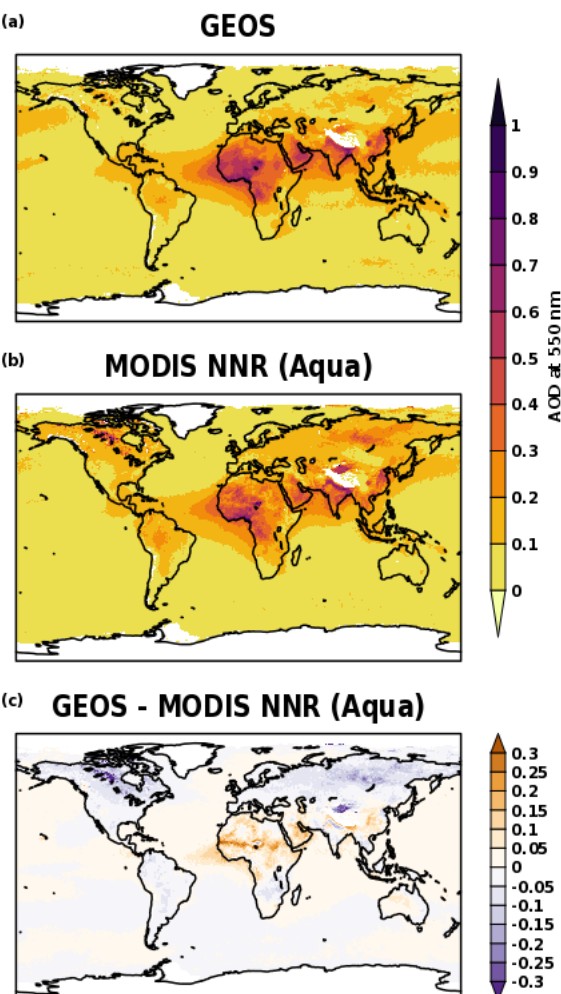

**Figure 4: Average AOD at 550 for the period of January 2016 through December 2019 in the (a) GEOS GOCART-2G benchmark simulation, (b) MODIS NNR observational product from Aqua, and (c) the difference between the model and observations.**






**Figure 5: Timeseries of ocean area-averaged (a) monthly mean AOD from the Aqua MODIS NNR observational product and the speciated AOD from the GEOS GOCART-2G benchmark simulation, (b) mean seasonal cycle, and the difference between the model and observations for the (c) monthly mean AOD and (d) seasonal cycle of AOD.**







**Figure 6: Timeseries of area-averaged monthly mean AOD from the Aqua MODIS NNR observational product and the speciated AOD from the GEOS GOCART-2G benchmark simulation over (a) North Africa, (b) South Africa, (c) Australia, (d) South Asia, (e) North America, (f) South America, (g) Siberia, and (h) Europe.**







**Figure 7: Timeseries of area-averaged monthly mean AOD from the Aqua MODIS NNR observational product and the speciated AOD from the GEOS GOCART-2G benchmark simulation over (a) the Iberian Peninsula, (b) Scandinavia, (c) the United Kingdom, and (d) central Europe.**



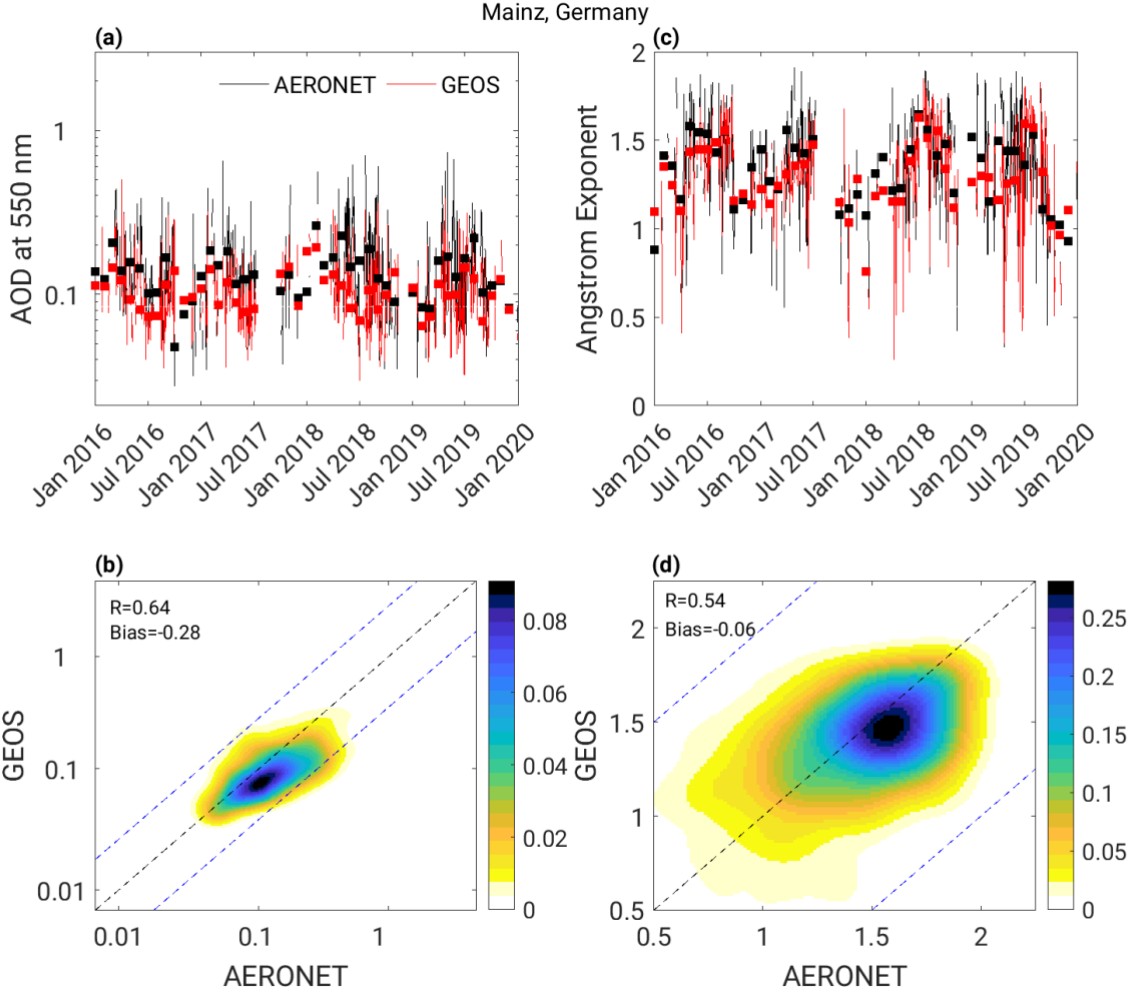

**Figure 8: (a) Timeseries of hourly AOD at 550 nm, (b) 2-D kernel density estimate for AOD at 550 nm computed as log(AOD+0.01), (c) timeseries of Angstrom exponent, and (d) (e) 2-D kernel density estimate for Angstrom exponent over the AERONET station in Mainz, Germany for all co-located data points from the observations and GEOS GOCART-2G benchmark simulation. The statistics in (b) are computed as log(AOD+0.01). The black dashed line in (b) and (d) indicates the one-to-one line with the blue dashed lines are the one-to-one line plus or minus one of the one-to-one line.**





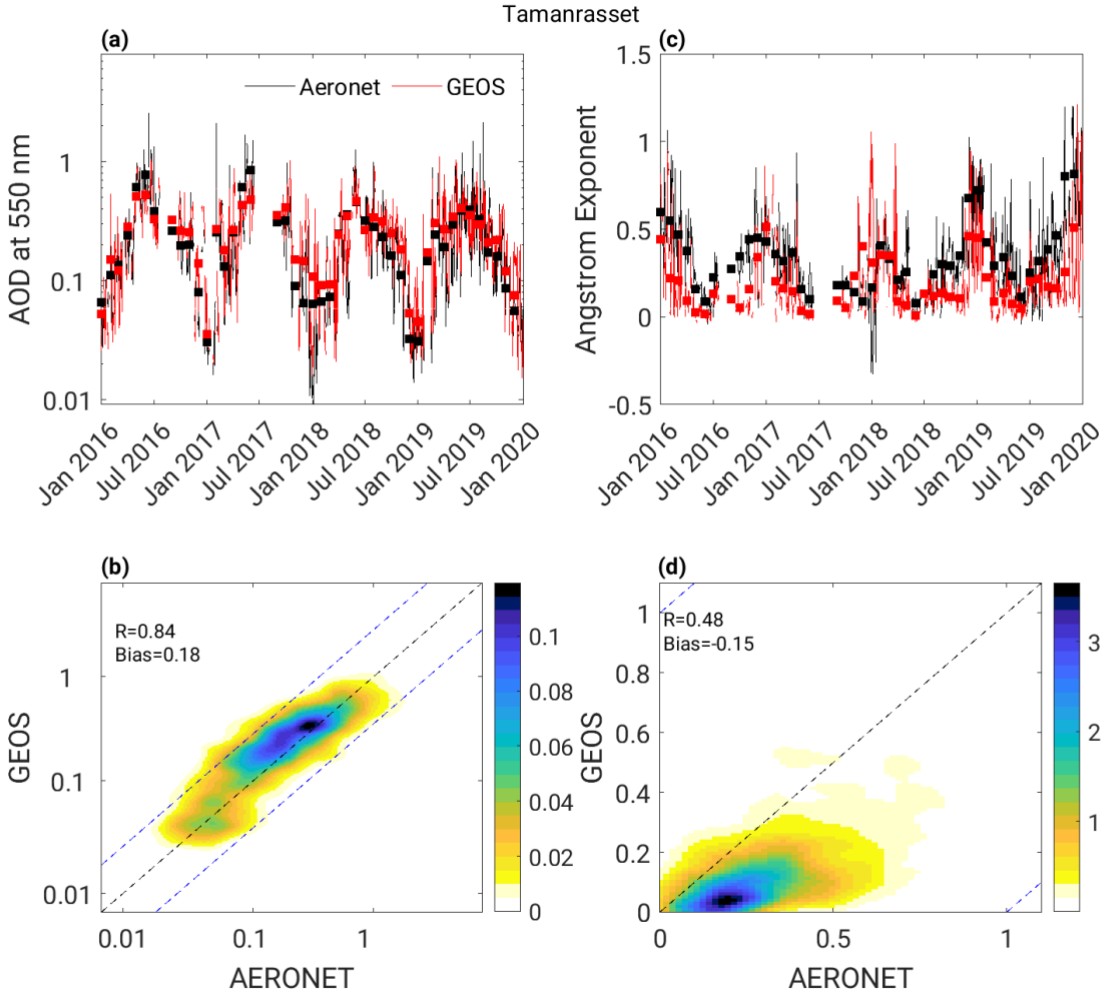

**Figure 9: (a) Timeseries of hourly AOD at 550 nm, (b) 2-D kernel density estimate for AOD at 550 nm computed as log(AOD+0.01),**
**(c) timeseries of Angstrom exponent, and (d) (e) 2-D kernel density estimate for Angstrom exponent over the AERONET station in**
**Tamanrasset, Algeria for all co-located data points from the observations and GEOS GOCART-2G benchmark simulation. The**
**statistics in (b) are computed as log(AOD+0.01). The black dashed line in (b) and (d) indicates the one-to-one line with the blue**
**dashed lines are the one-to-one line plus or minus one of the one-to-one line.**



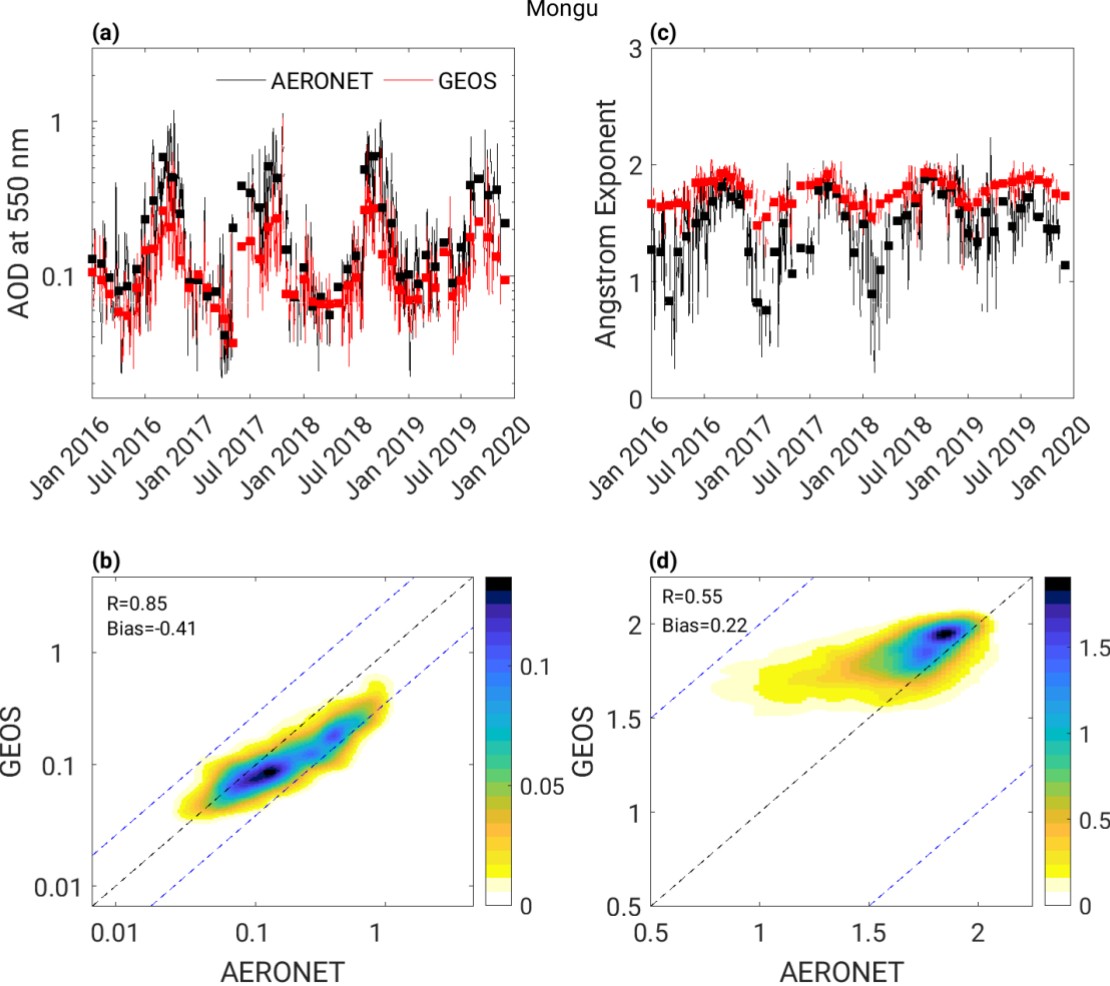

**Figure 10: (a) Timeseries of hourly AOD at 550 nm, (b) 2-D kernel density estimate for AOD at 550 nm computed as log(AOD+0.01), (c) timeseries of Angstrom exponent, and (d) (e) 2-D kernel density estimate for Angstrom exponent over the AERONET station in Mongu, Zambia for all co-located data points from the observations and GEOS GOCART-2G benchmark simulation. The statistics in (b) are computed as log(AOD+0.01). The black dashed line in (b) and (d) indicates the one-to-one line with the blue dashed lines are the one-to-one line plus or minus one of the one-to-one line.**







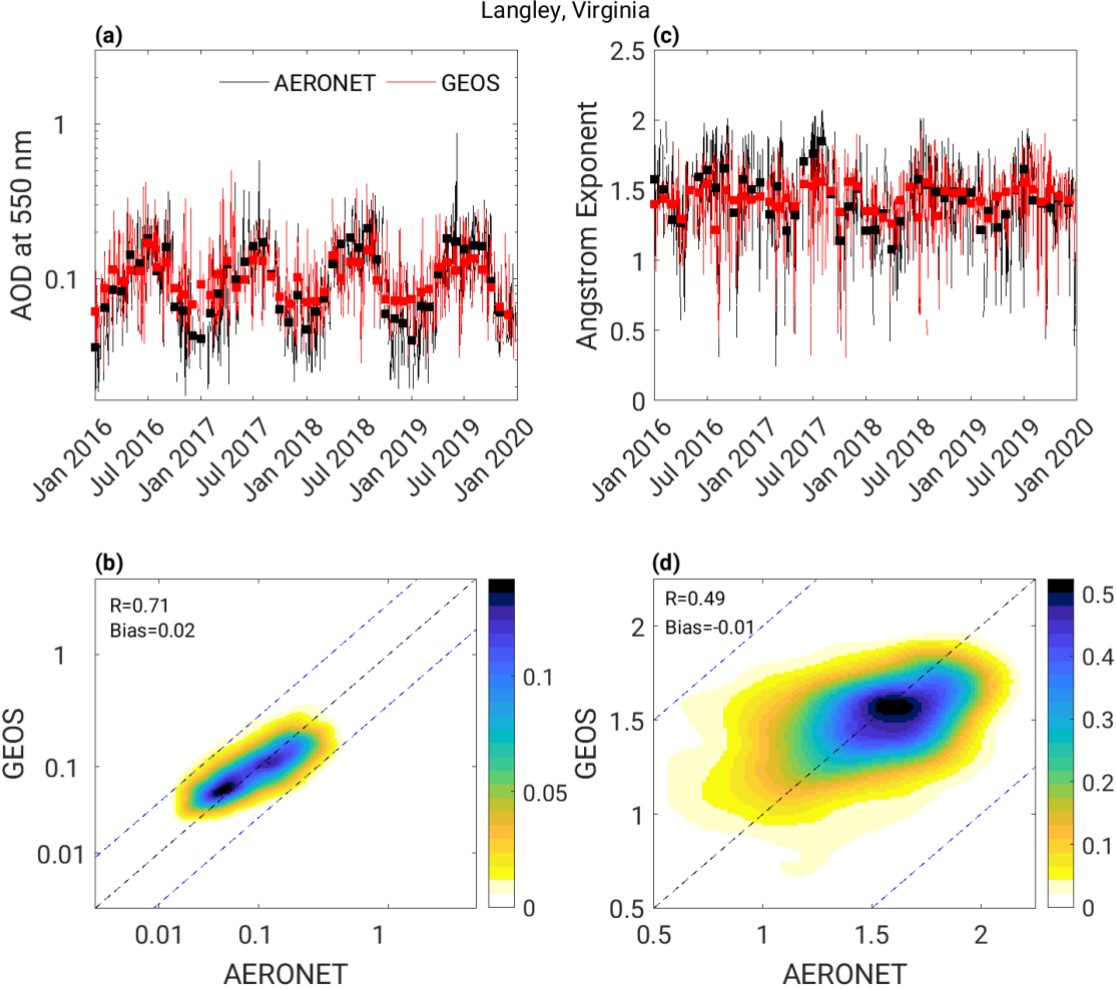

**Figure 11: (a) Timeseries of hourly AOD at 550 nm, (b) 2-D kernel density estimate for AOD at 550 nm computed as log(AOD+0.01), (c) timeseries of Angstrom exponent, and (d) (e) 2-D kernel density estimate for Angstrom exponent over the AERONET station in Langley, Virginia for all co-located data points from the observations and GEOS GOCART-2G benchmark simulation. The statistics in (b) are computed as log(AOD+0.01). The black dashed line in (b) and (d) indicates the one-to-one line with the blue dashed lines are the one-to-one line plus or minus one of the one-to-one line.**

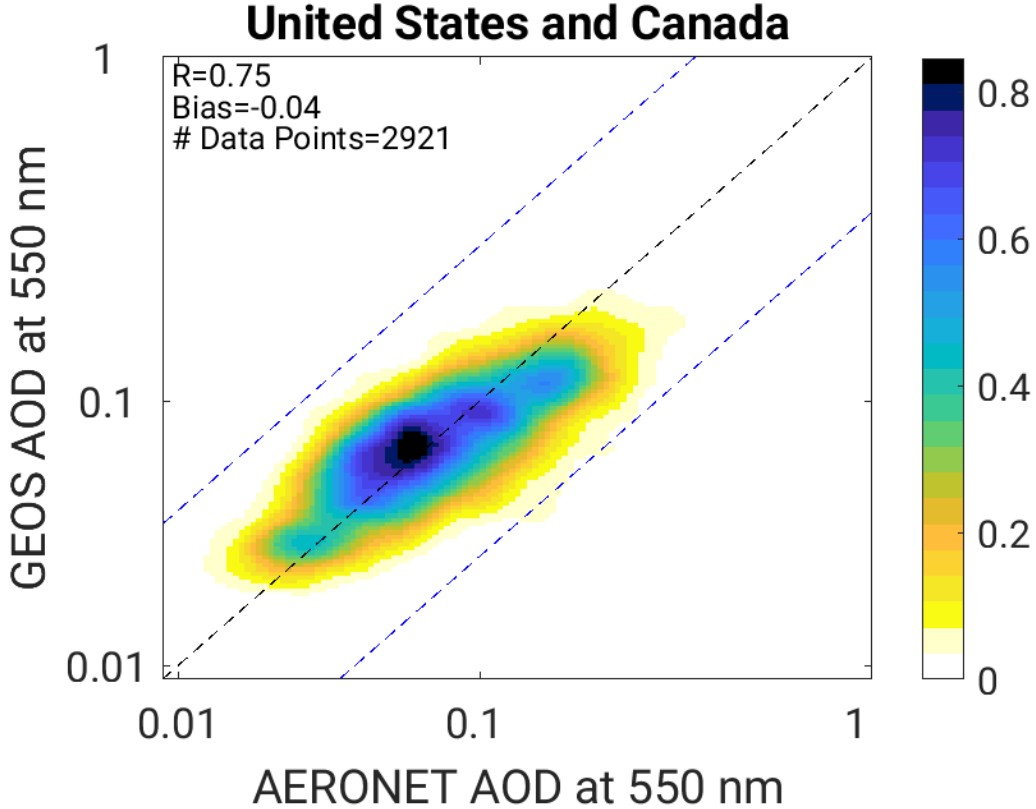

**Figure 12: 2-D kernel density estimate for AOD at 550 nm computed as log(AOD+0.01) from 77 AERONET stations across the**
**United States and Canada for co-located data points from the observations and the GOCART-2G benchmark simulation. The**
**statistics are computed as log(AOD+0.01). The black dashed line in (b) and (d) indicates the one-to-one line with the blue dashed**
**lines are the one-to-one line plus or minus one of the one-to-one line.**





A.

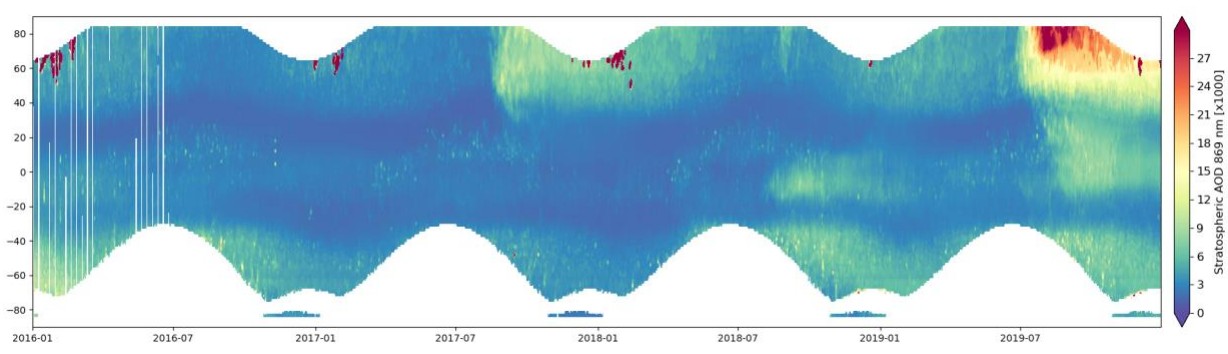


B.

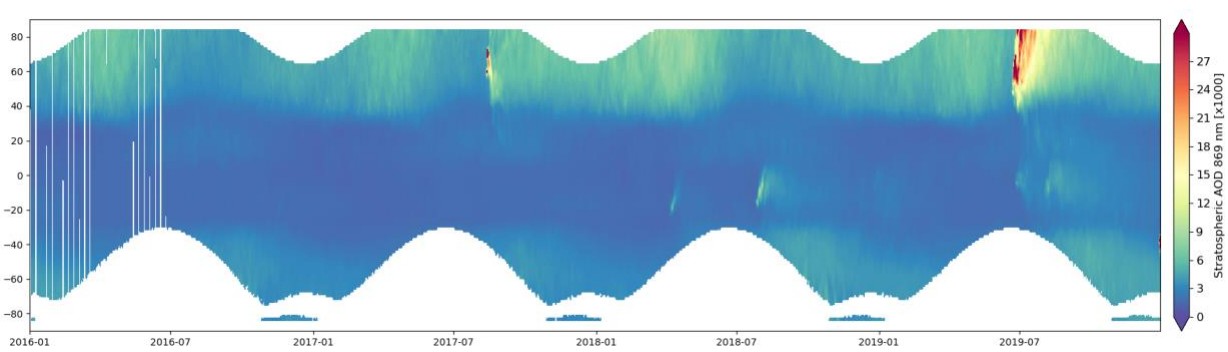

**Figure 13: Timeseries of zonal mean stratospheric AOD at 869 nm from (a) OMPS-LP observations and (b) the GEOS GOCART-2G benchmark simulation.**




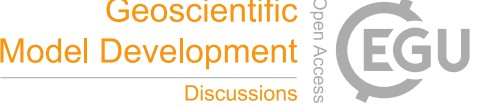


**Figure 14: Vertical profiles of total (aerosols + molecular) attenuated backscatter coefficient (km-1 sr-1) at 532 nm and derived from GEOS GOCART-2G simulations sampled on the CALIOP path and averaged over the continental United States, northern Africa (top row), South America and southern Africa (bottom row) for the period of June-July-August 2016. The solid lines are the median**
**of all profiles for CALIOP (black) and GEOS GOCART-2G (red). Shaded areas represent the 25th-75th percentile of all modelled and observed profiles.**



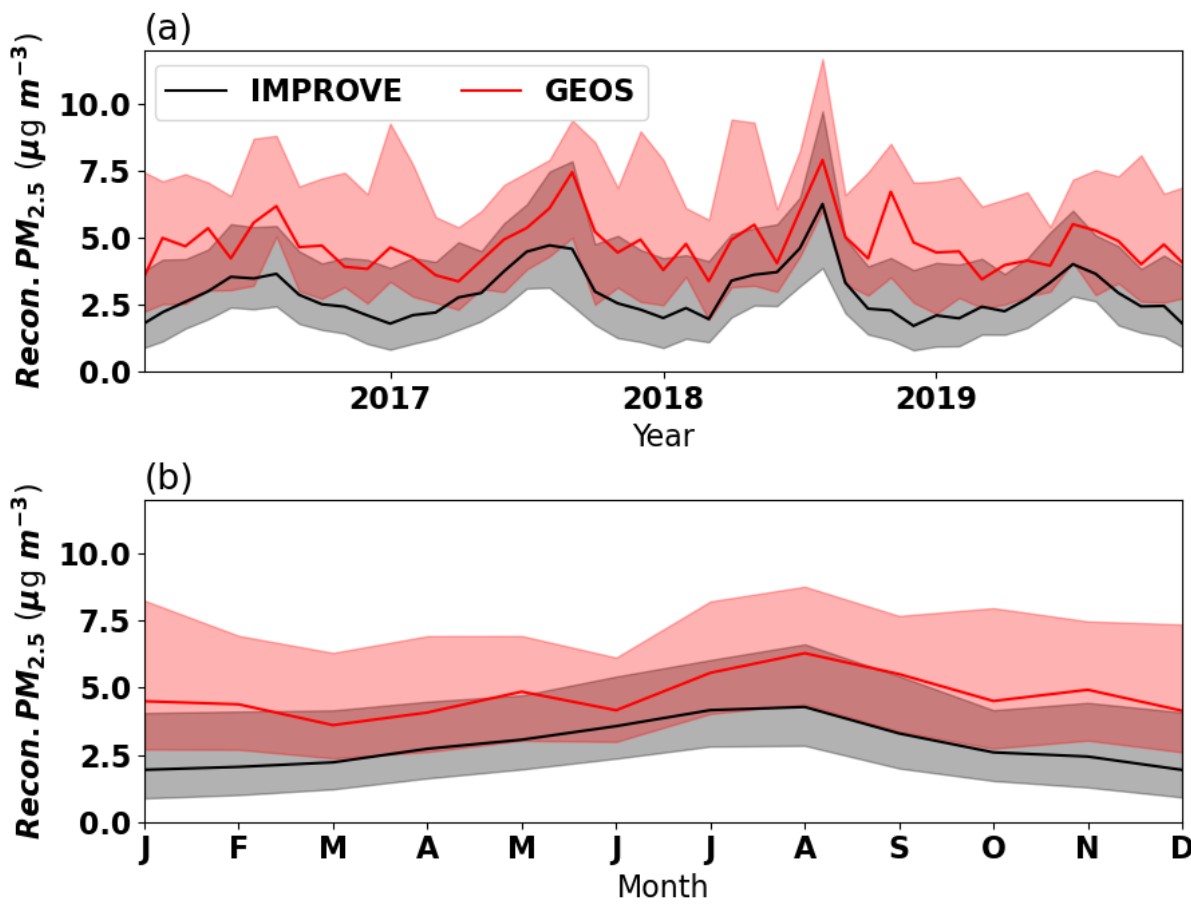

Figure 15: (a) Timeseries of monthly median and (b) median seasonal cycle of reconstructed $PM_{2.5}$ for the IMPROVE monitoring
stations across the United States from the observations and GEOS GOCART-2G benchmark simulation. Shading lies between the
$25^{th}$ and $75^{th}$ percentiles.



**Figure 16: Timeseries of the monthly median and median seasonal cycle for fine (a, e) sulphate, (b, f) nitrate, (c, g) organic carbon, and (d, h) dust averaged for the IMPROVE monitoring stations across the United States from the observations and GEOS GOCART-2G benchmark simulation. Shading lies between the 25[th] and 75[th] percentiles.**



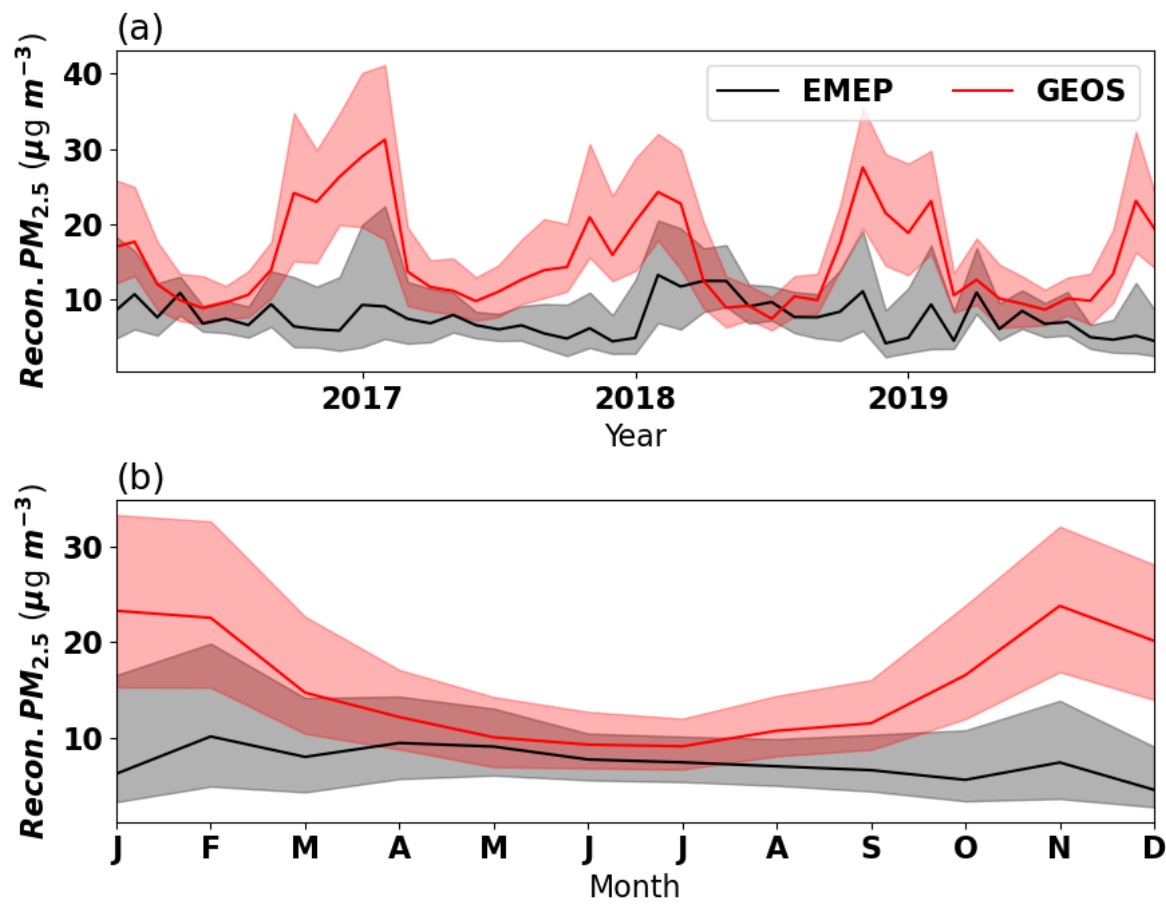

Figure 17: (a) Timeseries of monthly median and (b) median seasonal cycle of reconstructed PM$_{2.5}$ from four EMEP monitoring stations across Germany and one in Poland from the observations and GEOS GOCART-2G benchmark simulation. Shading lies between the 25$^{th}$ and 75$^{th}$ percentiles.






**Figure 18: Timeseries of the monthly median and median seasonal cycle for fine (a, e) sulphate, (b, f) nitrate, (c, g) organic carbon, and (d, h) black carbon for four EMEP stations across Germany and one in Poland from the observations and GEOS GOCART-2G benchmark simulation. Shading lies between the 25th and 75th percentiles.**


Appendix





**Table A1. Aerosol particle size ranges for dust, sea salt, carbon, sulphate, and nitrate in GOCART-2G. Note a lower and upper radius is not given for carbon or sulphate as there are not discrete size bins.**

| Aerosol Bin | Effective Radius Assumed for Radiation (µm) | Radius Lower Bound (µm) | Radius Upper Bound (µm) |
|---|---|---|---|
| DU001 | 0.636 | 0.1 | 1 |
| DU002 | 1.324 | 1 | 1.8 |
| DU003 | 2.301 | 1.8 | 3 |
| DU004 | 4.167 | 3 | 6 |
| DU005 | 7.671 | 6 | 10 |
| SS001 | 0.079 | 0.03 | 0.1 |
| SS002 | 0.316 | 0.1 | 0.5 |
| SS003 | 1.119 | 0.5 | 1.5 |
| SS004 | 2.818 | 1.5 | 5 |
| SS005 | 7.772 | 5 | 10 |
| BC | 0.0392 | | |
| BR | 0.0876 | | |
| OC | 0.0876 | | |
| SU | 0.156 | | |
| NI001 | 0.156 | | |
| NI002 | 2.10 | | |
| NI003 | 6.86 | | |


**Table A2. Optics table versions in the initial release of GOCART-2G**

| Specie | Optics Table |
|---|---|
| Black Carbon | optics_BC.v1_3.nc |
| Brown Carbon | optics_BRC.v1_5.nc |
| Dust | optics_DU.v15_3.nc |
| Nitrate | optics_NI.v2_5.nc |
| Organic Carbon | optics_OC.v1_3.nc |
| Sea Salt | optics_SS.v3_3.nc |
| Sulphate | optics_SU.v1_3.nc |