# Peer review of "Benchmarking GOCART-2G in the Goddard Earth Observing System (GEOS)"

_Geoscientific Model Development, 2023_

## Referee Comment (RC2)

In this study, the authors document the development effort for GOCART-2G model and evaluate the benchmark simulation against multiple observations. Overall, the manuscript is in good shape. I have a few comments for the authors to consider.

General Comments:

1. I would suggest that the authors show the changes or improvements of aerosol related fields, such as AOD, aerosol mass budgets, and aerosol mass concentrations, between GOCART-2G and previous GOCART version. Although the authors provide the documentation of changes in parameterizations and code refactoring, there is no information regarding how those changes impact actual fields (AOD, aerosol mass budgets, aerosol mass concentrations, etc). This is especially important for readers outside of the GEOS-GOCART community.

2. I think more effort is needed for the analysis of aerosol mass budget (i.e., section 4.1). It would be more helpful to show global mean total column mass, or burden in other words, and aerosol lifetime. I would suggest the authors compare them with previous GOCART and results from AeroCom III (e.g., Gliß et al., 2021) and other recent studies (e.g., dust from Kok et al., 2021). I think tables or bar plots showing the global mean statistics (e.g., burden, emission, deposition, chemistry production) are more helpful than the time series plots (Figure 2) and just showing the annual mean spatial distributions of emission and deposition from GOCART-2G (Figures 1 and 3).

3. It seems that the authors have focused on comparisons of AOD with MODIS and AERONET. However, I think some key metrics are either not shown or spread in many individual plots. I would suggest the authors improve the ways they present the results and add more discussions. First, tables showing statistics of global annual mean land, ocean, total AOD from GOCART-2G, previous version of GOCART, MODIS (Aqua and Tera), and MISR would be helpful. Figures 8-11 give limited information, most of which is similar to Figure 6. I would suggest the authors to combine them and show the key results. For example, just show panels (a) and (c) with statistics. The discussion of low AOD biases over Europe may need more effort. The comparison of surface $PM_{2.5}$ shows high biases for GOCART-2G. There is a low bias of attenuated backscatter between 1 km to 5 km over CONUS region. I would suggest the authors look at other seasons and annual mean as well.

Specific comments:

Lines 14-15, should be "the sources, sinks, and chemistry within".

Line 18, it is not clear to me. Do you mean "so that multiple instances of an aerosol …" or "such that …"

Line 138, could you give the full name for QFED when it is first referred to?

Line 151, should be "biogenic VOCs". Isoprene, monoterpene, and other terpenes are gas species.

Lines 158-159, I'm a bit confused here. Is OCS a prescribed species, a species with fixed concentration, or an active tracer in the model?

Lines 165-167, why the mechanism will not be used for typical model simulations.

Could you give any statistics to show the computational performance of GOCART-2G compared with previous version?

Do the authors perform any data sampling related to orbital space and time for comparisons with MODIS and CALIOP?

Line 275, is emissions for BC, OA, and sulfate from shipping included? I did not see carbonaceous emissions over ocean. If it is not included, what is the reason?

Figure 12 gives limited information and can be either put in supplement or combined with other plots.

Figure 14, I would suggest showing aerosol extinction. Why only show JJA results? What about annual mean and other seasons?

Reference:
Gliß, J., Mortier, A., Schulz, M., Andrews, E., Balkanski, Y., Bauer, S. E., Benedictow, A. M. K., Bian, H., Checa-Garcia, R., Chin, M., Ginoux, P., Griesfeller, J. J., Heckel, A., Kipling, Z., Kirkevåg, A., Kokkola, H., Laj, P., Le Sager, P., Lund, M. T., Lund Myhre, C., Matsui, H., Myhre, G., Neubauer, D., van Noije, T., North, P., Olivié, D. J. L., Rémy, S., Sogacheva, L., Takemura, T., Tsigaridis, K., and Tsyro, S. G.: AeroCom phase III multi-model evaluation of the aerosol life cycle and optical properties using ground- and space-based remote sensing as well as surface in situ observations, Atmos. Chem. Phys., 21, 87–128, https://doi.org/10.5194/acp-21-87-2021, 2021.

Kok, J. F., Adebiyi, A. A., Albani, S., Balkanski, Y., Checa-Garcia, R., Chin, M., Colarco, P. R., Hamilton, D. S., Huang, Y., Ito, A., Klose, M., Leung, D. M., Li, L., Mahowald, N. M., Miller, R. L., Obiso, V., Pérez García-Pando, C., Rocha-Lima, A., Wan, J. S., and Whicker, C. A.: Improved representation of the global dust cycle using observational constraints on dust properties and abundance, Atmos. Chem. Phys., 21, 8127–8167,

https://doi.org/10.5194/acp-21-8127-2021, 2021.

---

## Author Comment (AC1)

Collow et al. introduce and evaluate the GOCART-2G aerosol module based on a variety of observation-based data sets. The evaluation is informative and yields interesting results. But I think that especially the introduction, the conclusion section, and the abstract should be improved. Many key results are presented clearly in the figures and the main text, but they should also be summarized and discussed in the conclusion section. The summary section should ideally contain not simply a summary, but a synthesis of the results from the evaluation, for example with respect to results from the evaluation with MODIS and IMPROVE (compare my main comment #6 below). Ideally, the introduction and discussion should provide some context that may help the reader to better understand the results. Several key results could perhaps be discussed in the context of existing literature. A few points also deserve further discussion in the main text. AOD over the major source regions of anthropogenic aerosol in Europe, North America and South Asia in northern hemisphere spring and summer is lower in the GEOS-GOCART-2G model simulation compared to Aqua MODIS NNR data. Overall, the AOD difference between GEOS-GOCART-2G and Aqua MODIS NNR is smaller over South Asia compared to North America and Europe. There is also an interesting point about differences between Aqua MODIS NNR and Terra MODIS NNR, which I think could be further discussed in a few sentences, if possible in the light of existing literature. I do not suggest additional analysis within the framework of this manuscript. Instead, some of the suggestions that the authors made could be explained a bit better, and there could be additional attempts at synthesizing the findings. There should also be a better description of specific open issues that arise from the evaluation in the conclusion section, and perhaps the authors could mention more concrete ideas either for further investigation of these issues or for addressing them. The explanations of technical aspects require clarifications.

Thank you for the constructive review of our manuscript. The abstract, introduction, and discussion sections have been overhauled to better motivate the evaluation and synthesize the findings. Results from previous studies have been added to the introduction and the discussion now contains a bulleted list of priorities for future development in GOCART-2G.

Main comments:

1. Abstract: I recommend to cut all the details on code changes and instead summarize key results from the evaluation (see specific comments below). I appreciate the technical work that went into the code refactoring and the new features, and I understand how very important this type of model development is. But I find the information in the abstract hard to follow and possibly of limited value for the wider readership. I think it is enough to mention that the code has been refactored in the abstract and to explain some of the improvements in a revised section 2.3. The rest

should be left to documents such as user guides and more comprehensive technical documentation.

The code changes have been removed from the abstract as suggested. The abstract now contains additional details and results pertaining to the evaluation of GOCART-2G. The relevant text that was added is copied below.

"This MODIS-based analysis is corroborated by comparisons to MISR and selected AERONET stations, however discrepancies between the Aqua and Terra satellites indicate there is a diurnal component to biases in aerosol optical depth over South Asia and Northern Africa… Over Europe, GOCART-2G is unable to match the summertime peak in aerosol optical depth, opposing the observed late-fall and early-spring peaks in surface mass concentration. A comparison of the vertical profile of attenuated backscatter to observations from CALIPSO indicates the GEOS model is capable of capturing the vertical profile of aerosol however the mid-troposphere plumes of dust in the North Atlantic and smoke in the Southeast Atlantic are perhaps too low in altitude. The results presented highlight priorities for future development with GOCART-2G, including improvements for dust, biomass burning aerosols, and anthropogenic aerosols."

2. Introduction: I suggest to re-write the introduction to better motivate and explain the model development and the evaluation. In order to motivate the model development, I suggest to clearly explain in which contexts this new module will most likely be used, and which shortcomings motivated the additional development. I suggest to provide specific scientific background that prepares readers to better understand and appreciate the interesting results of the model evaluation. I find that at the moment, the introduction contains a lot of fairly general background information without a clear link to the results from this specific study. I recommend to use the introduction to motivate the model development steps described in the manuscript, to motivate why specific steps were taken for the model evaluation, and to prepare the readers for understanding the specific results. In case you added diagnostics or substantially revised them, this could also be mentioned and motivated. If you think this will help the reader, you could also explain and mention some open issues that your code refactoring addresses (although I recommend to focus on motivating the scientific aspects and the evaluation, even in case the evaluation used a standard package and did not include additional diagnostics compared to previous publications on GOCART). I think that instead of providing general and/or historical background, the introduction should serve to motivate and explain the rest of the manuscript. You could also motivate Section 4.2.3 on stratospheric AOD.

The introduction has been rewritten as suggested.

3. Please revise Lines 176-183 for clarity. Avoid jargon and explain advantages in order to motivate the changes. At the moment, it is not clear to me, how the sentence

starting in line 183 is linked to the preceding sentences. I think it is linked, but I am not sure.

The sentence starting on line 183 is not linked to the prior sentence and has therefore been moved to the start of a new paragraph.

4. The discussion section should put a much stronger focus on synthesizing and discussing the results from the evaluation.

New text has been added to the discussion section that focuses on interpreting the results.

5. Can you explain which model biases are inherited? Can you provide clues based on existing literature? It would be nice to point out where the results from this very informative and comprehensive evaluation add to existing knowledge and where they confirm previous insights. I think you could use the introduction section to summarize known issues and the conclusion section to point out where your evaluation has yielded new insights or may lead to new ideas.

The following lines have been added to the discussion section:

"Scientifically, no changes were made to dust, sea salt, nitrate, or sulphate when moving from the legacy GOCART code to GOCART-2G. Therefore, any biases in these species in GOCART-2G were inherited from prior versions of GOCART or introduced by changes in emissions."

Following the recommendation from other reviewers, figures and tables now quantify the differences between the two versions of GOCART so it discussed throughout if the biases were inherited.

6. Based on Fig.6, AOD biases for Europe, North America and South Asia, which are major source regions of anthropogenic aerosol, appear to all show a seasonal cycle, with smaller biases in winter. Fig. 16a for North America shows a seasonal cycle for sulphate in the IMPROVE data, but much less for GEOS-GOCART-2G. Do you have any thoughts on this and/or can you find information on this in the literature? Could the lack of a seasonal cycle in Fig. 16a be part of the reason for the seasonal cycle of the bias Fig. 6e? Is this a known issue?

The positive bias in surface sulfate with respect to the IMPROVE dataset is rather consistent throughout the year and is probably not behind the seasonal bias in AOD. There is a known low bias in the extinction for carbonaceous aerosol, which peaks during the summer months over the United States. Dust and nitrate contribute less than carbon to the total AOD over North America however could also be playing a role. As shown in panel h of Fig 16 (now 15), the surface mass concentration for dust is underestimated in the model during the summer months due to a lack of sources for local agricultural dust. The seasonal cycle for nitrate is also too large in magnitude. Text detailing these issues has been added to the discussion section.

And does it apply to Europe and South Asia as well? Figure 18a suggests that for Europe the answer might be yes, but that it is not limited to sulphate (as you noted in the text).

Europe and South Asia likely also suffer from the mass extinction efficiency bias for carbonaceous aerosol, however both regions have a larger mass loading from anthropogenic aerosol than the United States, and contributions from dust. There are likely multiple processes at play. Text has been added to the discussion section pertaining to Europe as well as a paragraph on the uncertainty in anthropogenic emissions. Regarding South Asia, the bias is only present in comparison to Aqua, indicating there is a diurnal component. An investigation of the diurnal cycle was included in the list of suggested areas for follow up.

Do you know whether this seasonal AOD bias is linked to a seasonal bias of precipitation?

Unfortunately, little has been documented regarding the precipitation in the version of GEOS that was used here and investigating that is beyond the scope of this study. While there are papers in the literature that look at precipitation in MERRA-2, convection and precipitation has changed dramatically since then in the GEOS model.

And/or is there a known seasonal bias in one of the source terms? Unless this is already understood, you could perhaps suggest to investigate potential links between biases in meteorological variables and biases in AOD and/or to investigate the seasonal cycle of source and sink terms in a follow-up study. This may or may not help to explain the seasonal cycle of the AOD bias.

Thank you for this suggestion. It has been added to the end of the discussion section.

All in all, I very much like the wealth of different diagnostics and how the authors present them. I find that the outcomes provide a very good motivation for this manuscript. But I would nevertheless like to encourage the authors to spend even more effort on trying to synthesize results in order to derive ideas and conclusions from their data analysis, and where this is useful, also to put their results into the context of the existing literature. Especially with respect to sulphate, it may also be interesting to speculate on potential effects of biases on ERFari+aci, although whether to include such speculation is a matter of taste and should be decided by the authors. I am not sure how relevant this aspect is for the GOCART applications.

Thank you for the valuable suggestions, which we feel ultimately led to an improvement in the manuscript. As a bulk aerosol module, GOCART is not ideal for studying aerosol-cloud

interactions, nor was the simulation run with two moment microphysics. For these reasons we chose not to speculate on potential biases in radiative forcing.

Specific and other general comments:

1. I suggest to write GEOS model instead of simply GEOS throughout the text. I suggest to also add the word model to the end of the title. Alternatively, the authors could consider writing "Goddard Earth Observing System model (GEOS)" each first time the acronym occurs in the title, the abstract and the main text. But I think that writing GEOS model would be much more accurate than GEOS when referring to the GEOS model, because GEOS obviously involves other activities apart from modeling or data assimilation.

GEOS is defined in the title, abstract, and main text. An effort was made to add "model" throughout the text.

Did you set up a case and evaluate it with existing diagnostics, add new diagnostics to a set of existing diagnostics, and/or are you also providing a new or an updated tool or framework for model diagnostics? If you actually provide a new or an updated tool or framework, you could make this visible by slightly changing the wording in the abstract and then explain it in the main text. Are the new or revised diagnostics part of the preprocessing software mentioned in the code availability section? If yes, you could for example stress this in the discussion section.

Apart from the stratospheric aerosol optical thickness and diagnostics specific to brown carbon, all other diagnostics were already present in the model. The new diagnostics are incorporated into the model source code itself, not preprocessing software.

Abstract:

Line 14f: ", which controls sources sinks and chemistry" -> sources and sinks of what? What about aerosol physics, deposition, wet deposition, etc.? I suggest to replace this statement by something like ", the aerosol component in the Goddard Earth System (GEOS) model" or ", an optional aerosol component in the Goddard Earth System (GEOS) model" or ", an optional aerosol component for the Goddard Earth System (GEOS) model", or simply ", which is part of the Goddard Earth System (GEOS) model". I included the GEOS because the acronym is used in the abstract but not explained.

This sentence has been updated to "The Goddard Chemistry Aerosol Radiation and Transport (GOCART) model, which controls the sources, sinks, and chemistry of aerosols within the Goddard Earth Observing System (GEOS), recently underwent a major refactoring and update to the representation of physical processes."

Line 16-19: The benchmark case is mentioned again in line 21. I suggest to shorten this because the rest is of little interest for the wider readership. You could replace "This paper ... From a science perspective, a" by "A".

This sentence has been removed from the abstract.

Line 16: I think that this is neither the right place for documenting code changes nor that the manuscript does a fair job at actually documenting these code changes. Perhaps, "outline" would have been a better word than "document". But I think this sentence should be cut.

This sentence has been removed as suggested.

I think you should mention additional key results from the model evaluation in the abstract

Additional evaluation results have been added to the abstract. Particularly, the lines below.

"… however, discrepancies between the Aqua and Terra satellites indicate there is a diurnal component to biases in aerosol optical depth over South Asia and Northern Africa...Over Europe, GOCART-2G is unable to match the summertime peak in aerosol optical depth, opposing the observed late-fall and early-spring peaks in surface mass concentration.  A comparison of the vertical profile of attenuated backscatter to observations from CALIPSO indicates the GEOS model is capable of capturing the vertical profile of aerosol however the mid-troposphere plumes of dust in the North Atlantic and smoke in the Southeast Atlantic are perhaps too low in altitude. The results presented highlight priorities for future development with GOCART-2G, including improvements for dust, biomass burning aerosols, and anthropogenic aerosols."

1. Introduction:

Line 31-34: I find this sentence confusing because the first part sounds like GOCART is used in a traditional ESM context while the second part does not mention coupled models.

A key feature of seasonal prediction, one of the examples mentioned in that sentence, is atmosphere-ocean coupling.

I suggest to directly mention the GOCART-2G aerosol module and explain in which GEOS model applications GOCART-2G will be used (data assimilation, forecasting, ...?) in order to motivate your study.

The first paragraph now introduces GOCART/GOCART-2G, details past uses, and concludes with the list of future GEOS applications ("GOCART-2G is intended to be used in future versions of GEOS numerical weather prediction, subseasonal to seasonal prediction, and reanalysis products, hence the need for proper documentation and evaluation.")

Optionally, you could also clarify whether GOCART is used for estimating ERFari+aci or for climate projections, whether it participates in CMIP and/or in AeroCom and/or whether there are plans to do so. Please also explain what setup you are using here and motivate this choice. I think the introduction should serve to motivate and explain this particular study.

The second paragraph now discusses the participation of GEOS-GOCART in AeroCom and ICAP. The GEOS model is not used for climate projections and therefore does not participate in CMIP.

The statement "[a]s general circulation models strive to take a comprehensive Earth-system approach" hints at traditional ESMs, and I am not sure this will be the main application for GOCART-2G. Instead, the authors could for example briefly provide some background on applications of aerosol reanalyses products.

While the benchmark evaluation presented here is performed in the context a meteorological replay to a reanalysis, GOCART is used in multiple configurations of the GEOS model. This includes the subseasonal to seasonal configuration of GEOS, which is characterized by coupling between the ocean and atmosphere. The planned development pathway for GEOS is to merge the NWP and S2S configurations such that numerous configurations of GEOS (including MERRA-3) are comprehensive Earth system models. Considering we do not assimilate aerosol optical depth in our benchmark simulation and that GOCART(-2G) is used in all operational configurations of GEOS at the present time, it is not appropriate to provide a background on applications of aerosol reanalysis products.

Line 35-52: This seems like a very general and somewhat arbitrary background on aerosol modules. I do not understand what some of these points have to do with your results and how this either helps to motivate your study or else how some of this scientific background helps readers to understand the results from your model evaluation. I suggest to focus on the issues which are most important for your evaluation and to explain them a fashion that ensures that this becomes clear.

This section of the introduction has been removed and replaced by a section that discusses results from previous evaluation papers.

Line 52-64: This sounds like a history of GOCART. I think that readers may instead be interested in what future applications you envisage for your module. Can and will this be used only for data assimilation or also climate projection, short term forecasts, etc.?

We feel it is important to honor the work done to develop legacy GOCART and show the variety of applications of the module. Hopefully it is now clearer that GOCART coupled to GEOS, is used for aerosol data assimilation, NWP forecasts, seasonal prediction, and reanalysis.

See also my point regarding lines 31-34 above. I suggest to revise the introduction to motivate and explain this study and to provide scientific background that prepares the reader for understanding the results.

The introduction has been reworked as suggested.

2 GOCART aerosol module in GEOS:

Line 166: Introduce GEOS FP

GEOS FP is introduced in the introduction.

Lines 176-183: Please try to explain advantages for the user. Please try to avoid expressions such as "multiple instances" or "child" or else explain them. Personally, I very well understand the meaning of the expressions "multiple instances" and "child" and I also understand your goals. But I still feel like I do not quite understand what you are actually trying to say because I lack the (model specific) background to link your goals and your technical explanation.

Two examples (copied below) were added to this paragraph to help explain, however this paragraph has been moved to the supplemental document based on reviewer suggestions.

"This means that the model is provided with characteristics of each carbon species, including optics, density, particle radius, and the fraction that enters as hydrophobic, and black, brown, and organic carbon utilize the same code to perform process-related calculations, thus eliminating duplicated code. An example of a passive instance that could be run using the same methodology would be to track and provide diagnostics for the portion of a species from a specific emission source, such as sulphate formed in response from volcanic emissions."

4 Evaluation of GOCART-2G:

Line 300: Please explain this point in some more detail, reminding readers of the different overpass times, and, and as far as possible, also try to interpret the differences.

The overpass times are now mentioned for Terra and Aqua.

Line 328: For South Asia, the difference between satellite and model AOD depends on whether Terra MODIS NNR or Aqua MODIS NNR is used, again suggesting the influence a diurnal cycle. Please explain this point in some more detail, and as far as possible, try to interpret the differences in the light of the different overpass times. Please consider linking your discussion to existing literature such as https://doi.org/10.5194/amt-11-4073-2018.

The diurnal cycle is mentioned in the text; however we are cautious about linking this to the existing literature. Levy et al. (2018) demonstrate that dark target from Terra is biased high with respect to Aqua and explicitly states, "users should not interpret global differences between Terra and Aqua aerosol products as representing a true diurnal signal." As shown in the figure below, MODIS NNR is the opposite of what is shown by Levy et al. (2018) – Terra is biased low with respect to Aqua over land. The difference between Terra and Aqua over the ocean is also much smaller than what is presented by Levy et al. (2018). A sentence has been added to the text noting the difference between the products. Schutgens et al. (2020) also show a comparison between Aqua and Terra however it is limited to the North Pole, which is not applicable for the regions in question (https://acp.copernicus.org/articles/20/12431/2020/acp-20-12431-2020.pdf).

[Figure]

Line 349: Could you please briefly elaborate on your comment regarding emissions from smaller scale sources?

This sentence now refers to uncertainties in the CEDS dataset and references McDuffie et al. (2020), which contains a discussion on all potential sources of uncertainty.

Line 449: Has planetary boundary height in the GEOS model version that is used here been evaluated with observations? If there is a reference, you could cite it.

To our knowledge, PBL height has not been evaluated in the model version we used.

5. Discussion:

Lines 455f: I suggest to omit this sentence, or else explain again what HEMCO is and what the advantage of this step is. I don't understand the meaning of "As part of the new species".

This sentence has been removed as suggested.

Line 457-459: You cloud simplify this by saying something like "we added brown carbon and simplified the addition of new species." At the moment, I have two other comments regarding these lines:

This sentence now states "Primary science changes focused on a repartitioning of carbonaceous aerosol, distinguished based on the emission sources for organic matter. "

Line 457: I understand your point. But this is the results section. And why do you repeat a point in line 457 that you have made in line 453?

The paragraph beginning on line 457 recaps the technical changes to the code. The inclusion of brown carbon, as specified on line 453, is a scientific change that was made to the model. These lines have been revised.

Line 459: Is there any physical explanation why ash should be an instance of dust?

The processes controlling ash (emission, advection, settling, and deposition) are like dust and would therefore require the same code.

Line 471-482: Please omit. This sounds like the introduction to another manuscript.

These lines have been removed from the discussion.

I encourage the authors to put a much stronger focus on synthesizing and discussing the results from the evaluation in the discussion section.

An effort has been made to improve the analysis by adding text to the discussion section on connecting the surface mass to the AOD over North America, the seasonal cycle of AOD over Europe, a discussion on the uncertainty of anthropogenic emissions, and a bulleted list of priorities for future evaluation and development of GOCART-2G.

Technical comments and suggestions:

Line 211: with -> in

Line 331: decent agreement -> decent agreement with respect to the annual cycle

(AOD is underestimated in all four regions in Fig. 7)

Line 411: GEOS -> GEOS-GOCART-2G

Line 432: do not assimilated -> did not assimilate?

All technical comments/suggestions have been addressed.

Reviewer 2
In this study, the authors document the development effort for GOCART-2G model and evaluate the benchmark simulation against multiple observations. Overall, the manuscript is in good shape. I have a few comments for the authors to consider.
 General Comments:
1. I would suggest that the authors show the changes or improvements of aerosol related fields, such as AOD, aerosol mass budgets, and aerosol mass concentrations, between GOCART-2G and previous GOCART version. Although the authors provide the documentation of changes in parameterizations and code refactoring, there is no information regarding how those changes impact actual fields (AOD, aerosol mass budgets, aerosol mass concentrations, etc). This is especially important for readers outside of the GEOS-GOCART community.

Since GOGART-2G was the culmination of years of work, we previously did not have an apples-to-apples comparison between legacy GOCART and GOCART-2G. We agree that this is an important comparison and have added results that compare the versions throughout the paper. However, due to computational resource concerns, we were only able to run the legacy GOCART simulation for one year.

2. I think more effort is needed for the analysis of aerosol mass budget (i.e., section 4.1). It would be more helpful to show global mean total column mass, or burden in other words, and aerosol lifetime. I would suggest the authors compare them with previous GOCART and results from AeroCom III (e.g., Gliß et al., 2021) and other recent studies (e.g., dust from Kok et al., 2021). I think tables or bar plots showing the global mean statistics (e.g., burden, emission, deposition, chemistry production) are more helpful than the time series plots (Figure 2) and just showing the annual mean spatial distributions of emission and deposition from GOCART-2G (Figures 1 and 3).

A Table has been added to show a comparison between aerosol burden, aerosol lifetime, and emissions/production for the year 2016 in the legacy version of GOCART and GOCART-2G as well as 2016-2019 for GOCART-2G. Comparisons are now made to Gliß et al. and Kok et al. where appropriate.

3.  It seems that the authors have focused on comparisons of AOD with MODIS and AERONET. However, I think some key metrics are either not shown or spread in many individual plots. I would suggest the authors improve the ways they present the results and add more discussions. First, tables showing statistics of global annual mean land, ocean, total AOD from GOCART-2G, previous version of GOCART, MODIS (Aqua and Tera), and MISR would be helpful. Figures 8-11 give limited information, most of which is similar to Figure 6. I would suggest the authors to combine them and show the key results. For example, just show panels (a) and (c) with statistics. The discussion of low AOD biases over Europe may need more effort. The comparison of surface PM2.5 shows high biases for GOCART-2G. There is a low bias of attenuated backscatter between 1 km to 5 km over CONUS region. I would suggest the authors look at other seasons and annual mean as well.

A Table has been added to document global annual mean land, ocean, and total AOD in the satellite datasets and both versions of GOCART as suggested. Additional text has been added to the discussion pertaining to the biases over Europe. Regarding the low bias of attenuated backscatter over CONUS, the sentence below has been added. Attenuated backscatter in other seasons is addressed in the response for Figure 14.

"GEOS-GOCART-2G overestimates attenuated backscatter near the surface and underestimates attenuated backscatter just above the boundary layer over the United States and South America regions, which may be due to insufficient convective transport between the boundary layer and free troposphere, or the lack of a plume rise parameterization for intense fires."

Specific comments:
 Lines 14-15, should be "the sources, sinks, and chemistry within".
This has been fixed.

Line 18, it is not clear to me. Do you mean "so that multiple instances of an aerosol …" or "such that …"
At the recommendation of Reviewer 1, this sentence has been removed from the abstract.

Line 138, could you give the full name for QFED when it is first referred to?
Quick Fire Emissions Dataset is now mentioned.

 Line 151, should be "biogenic VOCs". Isoprene, monoterpene, and other terpenes are gas species.
"Aerosols" has been replaced with "VOCs" as suggested.

Lines 158-159, I'm a bit confused here. Is OCS a prescribed species, a species with fixed concentration, or an active tracer in the model?

OCS is prescribed at the surface however it is an active tracer in the model as a fully interactive and advected species. The sentence in the text has been updated as shown below. Note this has been moved to the supplemental document.

"A tracer for carbonyl sulphide (OCS) is added to the model, with a prescribed surface mixing ratio boundary condition of 490 ppt$_v$ and is transported by the model such that chemistry can occur in the stratosphere."

Lines 165-167, why the mechanism will not be used for typical model simulations.
This mechanism is computationally expensive and would require time and resources that are not feasible in an operational setting. The sentence has been clarified to mention this.

Could you give any statistics to show the computational performance of GOCART-2G compared with previous version?
Sample timers for the legacy and GOCART-2G versions of the code have been added to the supplemental document.

Do the authors perform any data sampling related to orbital space and time for comparisons with MODIS and CALIOP?
Yes, for all comparisons to observations the model output is sampled according to the available observations.

Line 275, is emissions for BC, OA, and sulfate from shipping included? I did not see carbonaceous emissions over ocean. If it is not included, what is the reason?
Emissions for shipping are included. A smaller contour interval has been added to the figure to make this more evident.

Figure 12 gives limited information and can be either put in supplement or combined with other plots.
Figure 12 has been moved to the supplemental document as suggested.

Figure 14, I would suggest showing aerosol extinction. Why only show JJA results? What about annual mean and other seasons?
Aerosol extinction is a derived variable that requires an assumption for the extinction-to-backscatter ratios. We chose to show attenuated backscatter coefficient as that is a more direct measurement and reduces uncertainty in the model comparison. The JJA season was chosen as this is the season when aerosol tends to be maximized. While it is possible to extend the analysis to other seasons, this is a time-consuming process that likely would not impact our results (we instead chose to perform the additional simulation with legacy GOCART). A single JJA season has been used in other publications including Buchard et al. (2015) and Buchard et al. (2017).

Buchard, V. J., A. M. Da Silva, P. R. Colarco, et al. 2015. "Using the OMI aerosol index and absorption aerosol optical depth to evaluate the NASA MERRA Aerosol Reanalysis ." Atmos. Chem. Phys., 15: 5743-5760, doi:10.5194/acp-15-5743-2015.

Buchard, V., C. A. Randles, A. M. da Silva, et al. 2017. "The MERRA-2 Aerosol Reanalysis, 1980 -- onward, Part II: Evaluation and Case Studies." Journal of Climate, JCLI-D-16-0613.1, doi: 10.1175/jcli-d-16-0613.1.

Reviewer #3

This manuscript documents the GOCART-2G implementations, and a multi-year benchmarking simulation results in the Goddard Earth Observing System (GEOS) and compares the aerosol results with observations from multiple sources (e.g., satellite and ground networks). The main purpose is to provide a reference for this new aerosol package version. The paper is generally well-organized and clearly written. The scope of the study fits well with the GMD journal. In particular, I like the detailed descriptions of the model development, but some important information, such as comparisons with the previous version or peer model results, is surprisingly not included. Together with the relatively weak motivations, it is difficult to understand the significance of this new development and its implications to climate model development at large. I would suggest a major revision before publication.

Thank you for reviewing our manuscript. The introduction section has been revised to provide additional motivation and the model results for GOCART-2G are now compared against the previous version for the year 2016. Tables have also been added for easier comparison against other AeroCom models.

General comments:

1. Since this is the second generation of the Goddard Chemistry Aerosol Radiation and Transport (GOCART) model, it is a bit surprising that the authors did not show any comparison between the current and the previous versions. What are the improvements? Anything unfortunately gets worse? What are the driving factors for these changes? These are perhaps most important results to document in a paper like this. I strongly recommend these results be added in the revised version.

Since GOGART-2G was the culmination of years of work, we previously did not have an apples-to-apples comparison between legacy GOCART and GOCART-2G. We agree that this is an important comparison and have added results that compare the versions, particularly in the form of tables. However, due to computational resource concerns, we were only able to run the legacy GOCART simulation for one year.

2. As I mentioned above the details about the code changes and developments are highly appreciated. But a large fraction of these details seems belong to

supplementary. In the main body, it would be more useful to provide the rationales of why such model developments are needed. Is it because it helps reduce the model biases? Or it adds some important processes that were not represented in the previous generation?

The technical details have been moved to the supplemental document as suggested and a couple sentences have been added to explain the purpose of the changes (copied below).

"Three major changes with regards to aerosol speciation were implemented as part of GOCART-2G to either represent processes that were previously not included or improve the interaction between aerosols and radiation."

"A major refactoring of the GOCART source code was completed to improve performance, flexibility, and code quality within GOCART-2G. This was essential to allow for future development of the aerosol module and for the code to be effectively shared with external organizations."

3. More importantly, GOCART as a bulk aerosol scheme simplifies size distributions and lacks microphysics compared to modal and sectional schemes. What are the main reasons GOES continues not to update to those more advanced aerosol treatments? What are the trade-offs? Such information, if persuading, could motivate this study much better than the current version.

The following sentence has been added to the introduction section to explain why GOCART is still used within GEOS:

"Although GEOS is a modular system that can be run with other aerosol modules for research purposes (e.g., Case et al., 2023), sectional and modal schemes are too computationally expensive to be used in a near-real time, operational environment."

4. Extending from #3, how does this bulk aerosol scheme perform relative to peer models, like CMIP6? Comparing to other models is as important as to the observations in this context. I suggest the authors add other model results besides the previous GOCART results.

CMIP6 runs are formatted differently (free-running with equivalent emissions scenarios) from our benchmark simulation (constrained meteorology with a best estimate of daily emissions) and are therefore not appropriate to add into the tables or figures. It is beyond the scope of this paper to run GEOS according to the CMIP6 framework. Please note that GEOS is not intended for inclusion in CMIP6 or future CMIP comparisons currently. When applicable, comparisons have been made to past AeroCom results.

5. Many CMIP6 models appear to have too strong aerosol indirect forcing. This has become a major concern of the current model development. It would be helpful to include more details about how aerosol-cloud interactions are handled in this study and what is their aerosol indirect effect.

Our benchmark simulation does not consider aerosol indirect effects. This has been made clearer by adding the following sentence to the first paragraph of Section 4.

"The benchmark simulation uses a one moment microphysics scheme such that aerosols are not used as cloud condensation nuclei for the formation of liquid or ice clouds."

Minor comments:

L82: Glad that brown carbon is added. Can you show some results of the brown carbon bleaching effect? Is a 2-day e-folding decay good match with the observations?

Bleaching is not accounted for currently in GOCART-2G. The 2.5 day e-folding time is for the transition from hydrophobic to hydrophilic carbon.

L108: Change to $SO_4^{2-}$.

This has been fixed.

Table 1: It is good to know that GOCART-2G can run with different emissions sources and dataset resolutions. Can you briefly illustrate the choices in this table? Why not other sources?

The following sentence was added to justify the use of CEDS emissions.

"The CEDS emission data was chosen to be consistent with other modelling efforts including the Coupled Model Intercomparison Project Phase 6 (CMIP6; Feng et al., 2020) and the Chemistry-Climate Model Initiative."

Section 4.1: Add a table summarizing global budget, burden, lift times. Otherwise, it seems missing information under the title of "aerosol mass budget".

A table with the burden, life time, and emissions has been added to the section.

L324-325: Any results support this hypothesis that the negative bias is due to biomass burning aerosol?

Yes, the AERONET comparison for Mongu is representative of biomass burning aerosol and demonstrates an underestimate of AOD in GOCART-2G.

Section 4.2.2: do we need to consider the sampling differences between the coarse model grid spacing and the site observations? If so, any attempt to reduce this potential bias?

Sampling differences between the site observations and the model is a valid concern. The AERONET site located in Mainz, Germany has previously been shown to be representative of others in the area. Over the United States, we investigated 77 stations in total. Given that the vast majority demonstrated the same relationship between the model and the observations, we are not concerned about potential biases due to sampling. Figures for each of the individual sites can be found in the supplement.

Supplement: add a table to list all the input variables needed for another model to include GOCART-2G and/or the interface configurations. This will make this paper a better documentation for others trying to port GOCART-2G.

A table has been added to the supplement that lists the required imports.

Figure 8 capture: what do you mean by "the one-to-one line plus or minus one of the one-to-one line"? Isn't 1:1 minus 1:1 zero? Please clarify.

The one-to-one line has the formula y=x, while the other two lines can be represented as y=x+1 and y=x-1.

Figure 9: in panel (d), the two lines besides the 1:1 line are so far from the colored area. They are not useful at all. Perhaps, replace these lines with a standard deviation labelled with R and bias?

Due to the changes in the ranges for the x and y axes on the KDE plots for the AERONET stations, providing the same lines on each figure allows one to compare the biases between two different sites.

Figure 13: are there quality control (QC) flags in the OMPS-LP data? Have these QC flags been applied? Otherwise, limiting the comparison to where the satellite data are more reliable makes much more sense.

A QC filter has been applied based on scattering angle and the data has been cloud-cleared. Nevertheless, there is still a possibility for clouds to slip through.

---

## Author Response (AR2)

Line 16: "update to the representation of physical processes". I think it would be nice to replace this by something more specific, such as "updates, including updates of emissions and the addition of brown carbon."

This line has been updated to "The Goddard Chemistry Aerosol Radiation and Transport (GOCART) model, which controls the sources, sinks, and chemistry of aerosols within the Goddard Earth Observing System (GEOS), recently underwent a major refactoring and update, including a revision of the emissions datasets and the addition of brown carbon."

Line 21: "and underestimates aerosol extinction over northern hemisphere boreal forests, requiring further tuning of emissions." -> In case you mean boreal forests, please briefly mention boreal forests in the results section, e.g. in Sect. 4.2.1. In case you mean biomass burning regions, please change boreal forests to biomass burning regions. Only in case you mean boreal forests, I also recommend to rethink the "requiring further tuning of emissions". In this case, one idea might be to place "requiring further tuning of emissions" directly behind dusty regions in the previous line and perhaps to write "requiring further investigation" where it now says "requiring further tuning of emissions". Over boreal forests, not only emission tuning, but also missing SOA formation mechanisms might be an issue. Another option could be to replace "requiring further tuning of emissions" by "requiring further investigation and tuning of emissions". If you are going to change boreal forests to biomass burning regions, "requiring further tuning of emissions" would seem fine to me.

The end of this sentence has been modified to state "requiring further investigation and tuning of emissions".

Line 121: please explain "replay"

The word "replay" has been removed and the sentence was updated to "Precursor gases for sulphate and nitrate are prescribed based on a prior GEOS simulation that was coupled to the GMI stratosphere-troposphere chemical mechanism and constrained by MERRA-2 meteorology (MERRA-2 GMI; Strode et al., 2019)."

In the author response to the reviews, the authors cite Levy et al. (2018) stating "users should not interpret global differences between Terra and Aqua aerosol products as representing a true diurnal signal." I recommend to mention something like this also in the manuscript, perhaps somewhere around Line 302.

"However, Levy et al. (2018) points out that differences due to satellite should not be used to indicate diurnal biases." has been added to the end of the paragraph.

Line 311: I think the increase of global mean AOD in GOCART-2G moving the GOCART-

2G global mean closer to the MODIS global mean may in part be due to a newly introduced AOD overestimate in the Saharan outflow in GOCART-2G (Fig. 4d)? Please discuss briefly.

Yes, agreed. The following sentence has been added, "However, this could have resulted from an overestimate in the outflow of Saharan dust that was introduced in GOCART-2G (Figure 4)."

Line 339: 4.2.4 -> 4.2.5

Fixed.

Lines 331 to 339 and Figure 7: I would have perhaps expected not only dust but also sulphate to display a more pronounced annual cycle over Europe. Based on this, it may seem that wet deposition in summer might be overestimated and long- and medium-range transport of sulfate (and dust) underestimated for some mid-latitude continental regions. Figure 17 shows a small annual cycle for sulphate over selected stations in Germany (only) at the surface. I think that uncertainties related to wet deposition could be mentioned and perhaps be explored in a future study.

The following sentence has been added "Uncertainty remains pertaining to the wet deposition of aerosols and how that may contribute to biases in the seasonal cycle of AOD of Europe, warranting future investigation."

Lines 497 to 505: Given many other uncertainties, this seems very speculative. Das et al. investigate a single event. I do not think that their study alone provides sufficient evidence for this speculation here to be overly plausible.

An additional reference has been added that documented concerns with smoke transport to Europe in a GEOS simulation (https://doi.org/10.5194/egusphere-2023-1945).